



# Characterisation and quantification of organic carbon burial using a multiproxy approach in saltmarshes from Aotearoa New Zealand

Olga Albot[1,2], Joshua Ratcliffe[1,3,4] Richard Levy[1,5], Sebastian Naeher[5,6], Daniel J. King[6,7] Catherine Ginnane[5], Jocelyn Turnbull[5], Mary Jill Ira Banta[6], Christopher Wood[5,8], Jenny Dahl[5], Jannine Cooper[5],
Andy Phillips[5]

[1] Antarctic Research Centre, Victoria University of Wellington, Kelburn Campus, Wellington 6012, New Zealand
[2] The Nature Conservancy in Aotearoa New Zealand, 32 Salamanca Rd, Kelburn, Wellington 6012, New Zealand
[3] Department of Forest Ecology and Management, Swedish University of Agricultural Sciences, Umeå, Sweden
[4] Unit for Field-Based Forest Research, Swedish University of Agricultural Sciences, Vindeln, Sweden
[5] GNS Science, Gracefield, Lower Hutt, New Zealand
[6] School of Geography, Environment and Earth Sciences, Victoria University of Wellington, Kelburn Campus, Wellington 6012, New Zealand
[7] Wildland Consultants Ltd., Tower Junction Christchurch, PO Box 9276
[8] University of Arizona, 1200 E University Blvd., 85721 Tucson, Arizona, U.S.A.

*Correspondence to*: Olga Albot (olya.albot@vuw.ac.nz)

**Abstract.** Blue carbon ecosystems, such as saltmarshes, play a crucial role in sequestering atmospheric carbon dioxide by storing it as buried organic carbon, also known as blue carbon, for centuries to millennia. This has generated significant interest in restoring these ecosystems for climate change mitigation benefits. While international methodologies exist for generating blue carbon credits through coastal wetland restoration, their application in Aotearoa New Zealand is limited by a lack of data
on saltmarsh carbon stocks and accumulation rates. Additionally, to improve carbon mitigation estimates, research is needed to better understand the sources, composition and preservation of organic carbon in saltmarshes and the factors influencing its long-term storage. This study quantifies these metrics at five saltmarsh sites in Aotearoa New Zealand using 45 sediment cores analysed for elemental composition, stable isotopes, X-ray fluorescence (XRF), lipid biomarkers and Ramped-Pyrolysis Oxidation-Accelerator Mass Spectrometry (RPO-MS) in combination with Pyrolysis-Gas Chromatography-Mass
Spectrometry (Py-GC-MS). Results show high variability in soil organic matter properties, carbon stocks ($40.7 \pm 9.1$ to $112 \pm 100.3$ Mg C ha$^{-1}$), and accumulation rates ($0.56 \pm 0.23$ to $2.5 \pm 0.44$ Mg C ha$^{-1}$ yr$^{-1}$). An initial assessment indicates increased carbon accumulation following restoration at two sites. Stable isotope and lipid biomarker results show substantial contributions from saltmarsh vegetation to the organic carbon pool. Preliminary analysis suggests long-term preservation of plant-derived organic carbon in the oldest basal soil samples. The findings highlight the importance of accounting for spatial
variability within saltmarsh ecosystems in carbon assessments and underscore the need for further research to determine environmental factors influencing long-term carbon storage and preservation.

## 1 Introduction

Coastal wetlands, such as saltmarshes, mangroves and seagrass meadows, sequester atmospheric carbon dioxide ($CO_2$) and store it as buried organic carbon (OC; 'blue carbon') over centuries to millennia (Chmura et al., 2003; Mcleod et al., 2011).
Globally, these ecosystems are estimated to accumulate 53.65 Tg OC yr$^{-1}$, contributing 30% of carbon burial in ocean sediments (Wang et al., 2021). This storage potential has led to high public and private interest in protecting and restoring these ecosystems for their climate mitigation potential, and blue carbon credit methodologies have been developed for the voluntary carbon market (Friess et al., 2022; Lovelock et al., 2023; Needelman et al., 2018). As a result, research has increased on quantifying the carbon sequestration potential of blue carbon ecosystems (BCEs) over the past decade (Howard et al., 2023;
Macreadie et al., 2019). However, there remain major gaps in data required to characterise these systems, especially for temperate regions in the Southern Hemisphere (Bertram et al., 2021; Macreadie et al., 2021). Research is also required to better



understand the sources of OC accumulating in BCEs and the factors that influence its long-term preservation (Howard et al., 2023; Macreadie et al., 2019).

Saltmarshes occur at the interface between terrestrial, marine and estuarine settings and accumulate OC that is: a) produced
in-situ by saltmarsh plants (autochthonous sources) and b) derived from terrestrial and marine organisms that live outside the saltmarsh and are transported and deposited at the marsh surface by riverine runoff and tidal inundation (allochthonous sources; Howard et al., 2014; Middelburg et al., 1997). These OC sources often mix with siliciclastic minerals that are also transported via fluvial and coastal currents and are deposited to form minerogenic soils (Howard et al., 2014; Middelburg et al., 1997; Saintilan et al., 2013). Given the complex combination of autochthonous and allochthonous sediment contributions to the
below-ground carbon pool, distinguishing between these sources is critical to accurately understand the drivers of carbon sequestration and ensure that all sources are accounted for in the sequestration potential of blue carbon credit models.

A combination of factors controls the extent to which OC is preserved long-term following deposition and burial. Aerobic microbial respiration from oxygenated soil layers, as a result of disturbance or during low tides when the water table is low, as well as some anaerobic microbial processes (e.g., methanogenesis), lead to emissions of $CO_2$, methane ($CH_4$) and nitrous
oxide ($N_2O$), resulting in the loss of stored OC (Howard et al., 2023; Macreadie et al., 2013; McTigue et al., 2021). A suite of spatiotemporal (e.g., lateral fluxes of dissolved carbon), biogeochemical (e.g., microbial composition), biotic (e.g., benthic fauna), hydrological (e.g., inundation patterns), climatic (e.g., temperature) and anthropogenic (e.g., nutrient runoff) drivers also affect soil OC dynamics (Howard et al., 2023; Martinetto et al., 2016; 2023; Rosentreter et al., 2021; Russell et al., 2023; Watson et al., 2022; Xiao et al., 2024). Furthermore, the reactivity and turnover rates of autochthonous and allochthonous
sources can vary. Autochthonous OC in saltmarsh soils consists mainly of fresh organic matter (OM), which is labile and more readily available for microbial respiration (e.g., plant cellulose in aquatic macrophytes; Kaal et al., 2020; Komada et al., 2022). In contrast, allochthonous OC tends to be more resistant due to its recalcitrant nature (e.g., OM derived from woody plants) or physical protection by mineral association (Komada et al., 2022; Macreadie et al., 2025; Van De Broek et al., 2018). Improving our knowledge of sources and long-term fate of buried OC, both allochthonous and autochthonous, is essential to understanding
the carbon mitigation potential of saltmarshes and is increasingly important as these habitats face accelerated rates of sea-level rise and anthropogenic impacts (Macreadie et al., 2019; Spivak et al., 2019).

Interest in using BCEs to sequester carbon and offset anthropogenic emissions has increased in Aotearoa New Zealand (NZ) (Ministry for the Environment, 2022, 2024; Ross et al., 2024). A recent study estimating national soil carbon accumulation rates (CARs) for BCEs showed that the average CAR for three targeted saltmarshes is 0.89 Mg C ha$^{-1}$ yr$^{-1}$ (Bulmer et al.,
2024). This mean is significantly smaller than the global mean estimates for saltmarshes between 1.67 to 2.45 Mg C ha yr$^{-1}$ (Chmura et al., 2003; Ouyang & Lee, 2014; Wang et al., 2021). Only several studies have attempted to characterise the source of OC and its preservation characteristics in BCEs in NZ (e.g., Bulmer et al., 2020; Pérez et al., 2017; Sikes et al., 2009; Thomson et al., 2025). Further research is required to determine whether other saltmarsh sites across NZ bury carbon at rates closer to the global mean and resolve the large variability of OC sources and the long-term storage potential. Here we examine
five coastal saltmarsh systems at three locations that span 6.56° of latitude from Rangaunu Harbour in Northland (34 58' S, 173 13'E) to Pāuatahanui Wildlife Reserve in Wellington (41° 06' S, 174° 54' E). Our aim is to better characterise these saltmarsh systems, with three primary research objectives to (i) quantify the carbon stocks and accumulation rates across a range of saltmarsh habitats in NZ, (ii) assess the source and preservation characteristics of the buried OM, and (iii) produce a foundation synthesis to guide future research.

**2 Study sites**

Five saltmarsh sites in three geographic locations in the North Island of NZ were selected to capture a range of geomorphic and environmental settings.




### 2.1 Okatakata Islands, Omaia Island and Awanui, Rangaunu Harbour, Northland

Omaia and Okatakata Islands are situated within the 9,700-ha Rangaunu Harbour in Northland, which features tidal flats

colonised by mangroves and saltmarshes, and three main channels: Kaimaumau, Awanui, and Pukewhau (Fig 1; Heath et al., 1983). Stopbanks constructed around the southern shores and tributary streams since 1916 prevent spring tides and tidal surges from extending inland (Cathcart, 2005). Okatakata consists of two islands forming a 38-ha saltmarsh. Omaia is a 50-ha drained saltmarsh used as pastureland from 1937, with drains and stopbanks installed to prevent tidal flooding (Land Information New Zealand, 2018). Isolated patches of saltmarsh (Awanui) are present immediately south of Omaia. Historical imagery shows

that Awanui and Okatakata have remained undisturbed by human activities since at least the early 1940s.

### 2.2 Robert Findlay Wildlife Reserve, Pūkorokoro-Miranda, Waikato

Robert Findlay Wildlife Reserve (Robert Findlay) spans approximately 27 ha on the Firth of Thames' western coastline, within the 8,500-ha Pūkorokoro-Miranda coastal wetland (Fig. 1), which has been attributed to be a wetland of "international significance" (Gerbeaux, 2003). The wetland, located on a chenier plain, was converted to farmland in 1865, with the

installation of drains and a floodgate, and stilt ponds were created for shell extraction (Queen Elizabeth II National Trust, 1992; Woodley, 2016). By 1980, the drainage canals and flood gates were no longer maintained and gradually filled with silt, enabling the saltmarsh to regenerate. A conservation covenant was established in 1988 (Queen Elizabeth II National Trust, 1992).

### 2.3 Pāuatahanui Wildlife Reserve, Wellington

Pāuatahanui Wildlife Reserve (Pāuatahanui) is situated within the Pāuatahanui inlet, which forms the northern branch of the Te Awarua-o-Porirua Harbour, located 30 km north of central Wellington, and has extensive areas of saltmarsh approximately 50 ha in size (Fig. 1). The saltmarsh formed after the M8.2 earthquake on the Wairarapa fault in 1855, which caused widespread uplift around the Wellington region (Grapes and Downes, 1997; 2010). Historical records indicate that prior to this earthquake, the site of the present-day saltmarsh comprised subtidal or tidal flat environments (McManaway and Gaz, 1852; Park, 1841).

It has been suggested that 1855 marks the onset of marsh development, which had become well established by ca. 1865 (Stephenson, 1986). Sections of the saltmarsh were altered with drainage canals, and the area was used as pastureland for cattle and sheep until the 1980s (Sheehan, 1988). In 1980, Forest and Bird replanted four hectares of saltmarsh, and in 1984, the government established the Wildlife Management Reserve and began restoration on the remaining 46 hectares (Conwell, 2010; Guardians of Pāuatahanui Inlet, 2021).





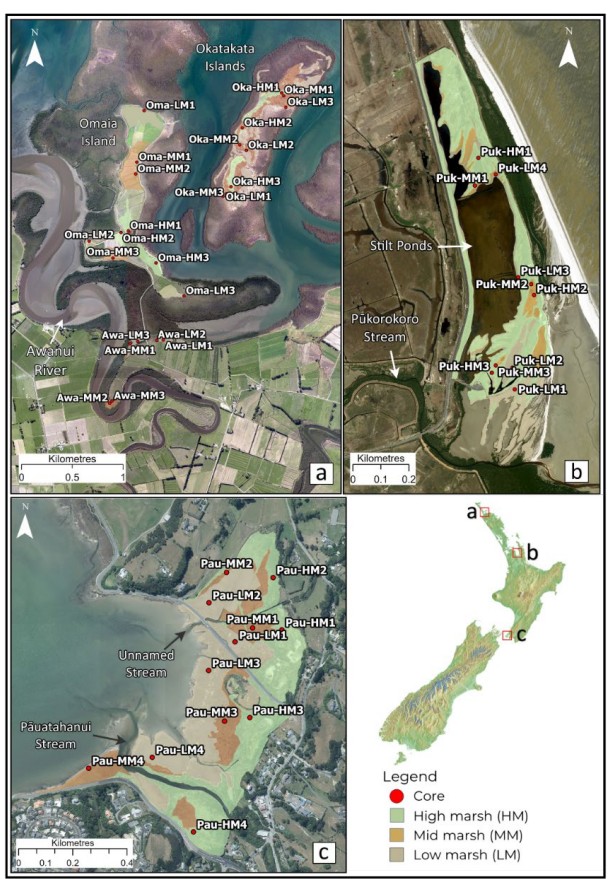

**Figure 1: Study sites and sample locations within a) Rangaunu Harbour, where Oma, Oka and Awa represent study sites Omaia Island, Okatakata Islands and Awanui, respectively; b) Robert Findlay Wildlife Reserve, where core locations are denoted as Puk; and c) Pāuatahanui Wildlife Reserve, where core locations are represented as Pau. Imagery is sourced from the LINZ Data Service and licensed for reuse under the CC BY 4.0 licence.**

## 3. Methods

### 3.1 Field sampling

Prior to sample collection, vegetation surveys were used to classify low, mid and high marsh zones based on saltmarsh plant community composition as reported in King (2022). For Omaia, historical aerial imagery was examined, and the former marsh zones were approximately delineated based on the observed general density of the vegetation, with higher density characterised as high and mid marsh and lower density zones as low marsh. Vegetation composition was described in 50 × 50 cm quadrats before the sediment cores were collected. The dominant vegetation types are summarised in Table S1 (Supplementary Materials).

Twelve sediment cores were collected from Pāuatahanui saltmarsh in November and December 2021. In January 2022, nine cores were collected from Omaia, nine cores from Okatakata, six cores from Awanui saltmarsh, and nine cores from the Robert Findlay. Fig. 1 and Table S1 show the sampling locations and details at the study sites.

All cores were collected with a gouge auger (6 cm diameter; 50 cm length), which recovers a cylindrical sediment core with minimal compaction (Smeaton et al., 2020). The cores were temporarily stored in PVC half-pipes with ice packs and transported and stored at 4°C at GNS Science, Lower Hutt, NZ. Before sub-sampling, the cores were described following the Troels-Smith (1955) sediment classification system. Following standard sub-sampling methodology as reported in Howard et



al. (2014), one core from each marsh zone from Omaia, Okatakata and Robert Findlay and two cores from each marsh zone from Pāuatahanui were sub-sampled in 2 cm depth increments from the top of the core down to 50 cm, and after that in 5 cm depth increments down to the base of the core. The rest of the cores were sub-sampled at 5 cm intervals between 0 and 20 cm and in 10 cm depth increments from 20 cm to the base. A total of 97, 25, 56, 75 and 61 samples were collected from Pāuatahanui, Robert Findlay, Awanui, Okatakata and Omaia cores, respectively. Where the base of the saltmarsh deposit was

ambiguous, soil samples were examined under the microscope for the presence vs absence of saltmarsh foraminifera to define the base depth of the marsh deposit following the methodology described in King et al. (2024).

## 3.2 Elemental and stable isotope analysis of Total Organic Carbon (TOC) and Total Nitrogen (TN)

Total organic carbon (TOC; wt%) and total nitrogen (TN; wt%) concentrations and their stable isotope composition ($\delta^{13}C_{org}$ and $\delta^{15}N$) were analysed in 314 samples at the Stable Isotope Laboratory at GNS Science in Lower Hutt, NZ. In brief, soil

samples were weighed before and after freeze-drying to calculate dry bulk density (DBD) following standard methodologies (e.g., Howard et al., 2014) and subsequently homogenised using a ball mill grinder. Large roots and aboveground biomass were removed. The samples were not size fractioned as this research focuses on bulk soil OC, which includes belowground living plant biomass (e.g., small rootlets and rhizomes; Macreadie et al., 2017). TOC and $\delta^{13}C_{org}$ were determined on acidified samples (treated with 10% HCl for 12 hours) by elemental analysis isotope ratio mass spectrometry (EA-irMS) using an

EuroEA3000 model. TN and $\delta^{15}N$ values were analysed on untreated samples (Smeaton et al., 2024; Sollins et al., 1999). Internal reference standards for $\delta^{13}C_{org}$ (Cane Sugar -10.3‰, beet sugar -24.6‰ and EDTA -31.1‰) and $\delta^{15}N$ (Leucine 2.0‰, EDTA 0.58‰ and Caffeine -7.8‰) were run every 10 samples. $\delta^{13}C_{org}$ and $\delta^{15}N$ values are reported in permil (‰) relative to the Vienna Pee Dee Belemnite (VPDB) standard and AIR, respectively. The C:N ratio is reported as the molar ratio of TOC to TN. The analytical precision of the measurements is ±0.2 wt% for TOC, ±0.1 wt% for TN, ±0.2‰ for $\delta^{13}C$ and ±0.3‰ for

$\delta^{15}N$.

## 3.3 Chronology

Robust age-depth models for the cores are required to accurately calculate CARs. Here we use a published age model for Pāuatahanui (King et al., 2024) that was developed using a Bayesian framework in the R package *rplum* (Blaauw et al., 2024). The age model assumes the base of the Pāuatahanui saltmarsh formed in 1855, when uplift during a Mw 8.2 earthquake on the

Wairarapa fault created a suitable platform on which the saltmarsh established (Grapes & Downes, 1997; Stephenson, 1986). A layer of shelly sands and clays at the base of the cores records the 1855 uplift event. The age-depth model developed by King et al. (2024) was used to interpolate age-depth estimates for several cores. Core Pau-HM3 (-41.1021537, 174.9157384), collected in close proximity to core PauM1 from King et al. (2024) (-41.1023944, 174.9155194), provided carbon accumulation rates. Additionally, core Pau-HM1 was interpolated to estimate ages for samples with biomarker data.

Lead isotope data were used to produce new age-depth models for one core selected from three locations at the other two study sites: Okatakata (Oka-MM1), Awanui (Awa-MM2) and Robert Findlay (Puk-MM1). Each core was sub-sampled in 1-cm increments, and the number of samples varied from 9 to 16 based on the core length and the stratigraphy. Lead isotope data were generated using gamma and alpha spectrometry conducted at the radio-isotope facility at the Institute of Environmental Science and Research (ESR), Christchurch, NZ. All samples were measured with gamma spectrometry and every second

sample in each core was measured with alpha spectrometry. For gamma spectrometry, sediment samples were packed into petri dishes and left to equilibrate for three weeks and then analysed to detect radionuclide activity to include $^{210}Pb$, $^{137}Cs$, $^{228}Ra$ and $^{226}Ra$ (Arias-Ortiz et al., 2018; Goldstein & Stirling, 2003). For alpha spectrometry, the samples were first processed to prepare the granddaughter $^{210}Po$ source, and the activity of $^{210}Po$ was then measured to calculate excess $^{210}Pb$ activities. Decay of excess $^{210}Pb$ activity (half-life = 22.5 years; Appleby, 1998; Arias-Ortiz et al., 2018) and $^{137}Cs$ discharge peak in

~1965 (Goff & Chagué-Goff, 1999; King et al., 2024) were used to determine the rate of sediment accumulation. Age-depth



models were generated using *rplum* (Blaauw et al., 2024). Unlike other $^{210}$Pb models, *rplum* does not require a bottom equilibrium depth to represent 'background $^{210}$Pb', allows for ad-hoc sample selection so the entire core does not need sampling, and enables the inclusion of secondary chronological data (e.g., $^{137}$Cs and radiocarbon dates).

$^{14}$C dating was attempted on sieved sedge and rush fragments (>1 mm) from basal samples from cores Oka-MM1, Awa-MM2 and Oma-MM3 at the Rafter Radiocarbon Laboratory at GNS Science, Lower Hutt, NZ. However, the reported calibrated ages for all basal samples were modern (post-bomb), and therefore unsuitable for inclusion in the age-depth models and for estimating a date of former saltmarsh establishment at Omaia.

### 3.4 Carbon stocks and accumulation rates

DBD (g cm$^{-3}$) was calculated by dividing the mass of the dry sample by the sample volume prior to drying (Howard et al., 2014). Organic carbon density (CD; g C cm$^{-3}$) was calculated by multiplying bulk density by the OC content for each depth interval (wt%; Howard et al., 2014). TOC stocks (Mg C ha$^{-1}$) for each core were calculated by integrating the depth intervals (2, 5 or 10 cm) over the depth range of the core. CARs for each study site were calculated by dividing the TOC stock for each depth interval by the corresponding age as per the *rplum* age-depth models. Mean CARs were then calculated over the entire marsh core.

### 3.5 X-ray fluorescence (XRF)

Elemental abundances in 314 samples were measured using an Olympus Vanta M-series XRF portable scanner at Victoria University of Wellington, Wellington, NZ. The scanner has a nine-mm-diameter primary beam connected to a workstation that allows remote operation. Approximately 2-3 g of dried and homogenised sediment was placed in plastic tubes (5 ml; 15 mm diameter), resulting in sediment thickness in the tube of >10 mm. Each sample was measured using the standard 'Geochem 3-Beam' method (50, 40 and 10 kV beams set for 30, 30 and 40 seconds 'live time' respectively) built into the scanner, which is designed to detect and quantify major and some trace elements, including Al, Ca, Fe, K, Si, Mn, S, Sr, Ti, Zn, Cu, Pb, Zr, Nb and Rb. Calibration measurements were taken at the start and end of each analytical period for nine external standards provided by the United States Geological Survey (USGS; AGV-2 Andesite; BHVO-2 Hawaiian Basalt; COQ-1 Carbonatite; W-2 Diabse; SGR-1 Green River Shale; SCo-1 Cody Shale) and the Geological Survey of Japan (GSJ; JR-2 Igneous; JG-2 Igneous; JF-2 Igneous). Results are reported as absolute concentrations in parts per million (ppm).

### 3.6 Organic carbon fingerprinting

### 3.6.1 Lipid extraction and Gas Chromatography-Mass Spectrometry (GC-MS) analysis

Selected samples from Pāuatahanui (Pau-HM1 n=5, Pau-MM4 n=6, Pau-LM4 n=6), Okatakata (Oka-HM2 n=5, Oka-MM1 n=5, Oka-LM1 n=6) and Awanui (Awa-MM2 n=9) were analysed to determine their lipid biomarker compositions. Samples were selected along the core profiles at intervals where pronounced shifts or distinct changes were observed in TOC contents, C:N ratios, $\delta^{13}C_{org}$ and $\delta^{15}N$ trends. Analyses were carried out in the GNS/VUW Organic Geochemistry Laboratory at GNS Science, Lower Hutt, NZ, following the methodology described in Naeher et al. (2012, 2014) with some modifications. In brief, 3-6 g of freeze-dried, homogenised sediment was extracted (×4) by ultrasonic extraction using dichloromethane (DCM)/methanol (MeOH) (3:1, v:v) for 20 minutes each time. An internal standard consisting of 5α-cholestane, *n*-C$_{19}$ alcohol and *n*-C$_{19:0}$ fatty acid was added to the total lipid extracts (TLEs) for quantification. Elemental sulphur was removed by adding activated copper. TLEs were separated into apolar (F1) and polar (F2) fractions by silica gel chromatography using *n*-hexane and DCM/MeOH (1:1), respectively. Before GC-MS analysis, polar fractions were derivatised with BSTFA in pyridine at 80°C for one hour. The resulting lipid fractions were analysed by GC-MS on an Agilent 7890A GC system equipped with an Agilent J&W DB-5MS capillary column (60 m × 0.25 inner diameter × 0.25 μm film thickness) and with a splitter coupled to an Agilent 5975C inert MSD mass spectrometer and flame ionisation detector (FID). The oven temperature programme on the



GC-MS started at 70°C, maintained for 1 minute, and then increased to 100°C at a rate of 20°C min[-1]. Then, the oven was heated to 320°C at a rate of 4°C min[-1] and kept isothermal for 20 minutes. Helium was used as carrier gas, maintaining a constant flow rate of 1.0 mL min[-1]. Samples (1 μL) were injected spitless at an inlet temperature of 300°C. The mass spectrometer was operated in full scan ($m/z$ 50–700) mode with an electron impact ionisation at 70 eV, with a source temperature of 230°C.

The GC-MS data, focused on $n$-alkanes present in the F1 fractions and steroids detected in the F2 fractions, were interpreted with Agilent MassHunter/Chemstation software based on relative retention times and diagnostic mass spectra. Several indices based on $n$-alkane distributions, such as the carbon preference index (CPI), odd-over-even predominance (OEP), the average chain length ratio (ACL), and the relative contribution of aquatic plants relative to terrestrial biomass ($P_{aq}$ index), as well as $C_{28}$ and $C_{29}$ stanol-sterol ratios were calculated as described in Section 4.8.

### 3.6.2 Ramped-Pyrolysis Oxidation-Accelerator Mass Spectrometry (RPO-AMS)

RPO enables thermochemical separation of complex particle-associated carbon mixtures, which are oxidised to $CO_2$ and then graphitised for radiocarbon measurement. Labile, syn-depositional carbon is generally found in low-temperature pyrolysis fractions and older, recalcitrant carbon components are found in the higher-temperature fractions (Ginnane et al., 2024; Rosenheim et al., 2008). Radiocarbon content is typically measured on four to seven temperature fractions to establish an age profile of the composite carbon and identify a plateau of younger ages within the low-temperature range, which provides the most accurate representation of the depositional age (Ginnane et al., 2024; Rosenheim et al., 2008).

RPO-AMS analysis was run at the Rafter Radiocarbon Laboratory at GNS Science, Lower Hutt, NZ, following the methodology described in Ginnane et al. (2024), with minor modifications (i.e., sieving to obtain a homogenous, non-biased sample). These analyses were performed on one basal sample each from Okatakata (Oka-MM1, 28-30 cm depth) and Awanui (Awa-MM2, 90-95 cm depth). In brief, up to 5 g of each sample was sieved into <90 μm fractions to avoid macro-organic matter (e.g., single plant species-derived fragments) that would tend to bias the average sedimentary OM composition. Samples were homogenised, freeze-dried, and acid-washed with 1M HCl for 16 hours, rinsed to neutral, and freeze-dried again. Samples were then loaded in a pyrolysis reactor, and the pyrolysis furnace heated from room temperature to 700°C at a rate of 5°C min[-1], with helium as the carrier gas at a constant flow rate of 35 mL min[-1]. The volatilised carbon compounds were then oxidised to $CO_2$ (gas) with $O_2$/He (4 mL min[-1] and 7 mL min[-1] respectively) over platinum, nickel, copper, and silver wires, and the product gas was passed through the LI-COR Li-820 detector, producing a thermograph of $CO_2$ evolution with time. The shape of the thermograph reflects the presence of multiple carbon components within the OC mixture. Inflection points in the thermograph can serve as a first-order approximation of temperature intervals that should be sampled to separate these components. Water is removed from the sample gas, and the $CO_2$ is cryogenically trapped in a series of discrete stainless-steel traps according to split temperatures, resulting in multiple $CO_2$ aliquots partitioned by temperature range. The trapped $CO_2$ is then quantified, sealed into a Pyrex tube with pre-combusted CuO and Ag wire, recombusted at 500 °C for 4 hours, and graphitised for radiocarbon measurement (Ginnane et al., 2024; Turnbull et al., 2015). $\Delta^{14}C$ values and conventional radiocarbon ages (CRAs) are reported as defined by Stuiver & Polach (1977) and fraction modern ($F_m$) values are reported as defined by Donahue et al. (1990).

The $\Delta^{14}C$ data for each pyrolytic split is used to calculate the relative proportion of syndepositional (i.e., modern) OC versus recalcitrant carbon (i.e., older/reworked carbon) based on the modified isotopic mixing model by Broz et al. (2023) provided in Equation 1, below.

$$C_{modern} = TOC \left( \frac{\Delta^{14}C_{split} - \Delta^{14}C_{last\_split}}{\Delta^{14}C_{modern} - \Delta^{14}C_{last\_split}} \right) \qquad (1)$$

where $C_{modern}$ is the modelled fraction of syndepositional carbon, TOC is the total OC content (wt%) of the bulk sample, $\Delta^{14}C_{split}$ is the measured $\Delta^{14}C$ value of each pyrolytic split, $\Delta^{14}C_{modern}$ is a typical value for a modern post-bomb OC endmember (where



$\Delta^{14}C$ is assumed to be 0‰ in materials deposited in the last 2 kyr) and $\Delta^{14}C_{last\_split}$ is the final $\Delta^{14}C$ pyrolytic split value where all labile carbon components are considered to be degraded based on the Pyrolysis-Gas Chromatography-Mass Spectrometry (Py-GC-MS) composition results as described below.

### 3.6.3 Pyrolysis-Gas Chromatography-Mass Spectrometry (Py-GC-MS)

Py-GC-MS analyses imitate the ramped pyrolysis process without oxidation to determine the molecular distributions that correspond to the radiocarbon measurements, which provide information about the origin, relative quantities, and degradation states of different OM sources.

Py-GC-MS analysis was undertaken in the GNS/VUW Organic Geochemistry Laboratory at GNS Science, Lower Hutt, NZ, following Ginnane et al. (2024). These analyses were performed on one basal sample each from Okatakata (Oka-MM1, 28-30 cm depth) and Awanui (Awa-MM2, 90-95 cm depth). In brief, acid-treated sediment was analysed in a microfurnace-type pyrolyser, Frontier Lab PY-2020iD Double-Shot Pyrolyser, equipped with a Frontier Lab MJT-1030E Microjet Cryo-Trap and a Frontier AS-1020E Auto-Shot Sampler. Using the same temperature steps resulting from RPO analysis of the samples, the successive ramps were run in thermal desorption mode at 10°C min$^{-1}$. The higher temperature fractions were obtained iteratively by reinserting the same sample cup and gradually increasing the temperature. The evolved compounds (pyrolysate) were captured in a cryo-trap maintained at $-190$ °C using liquid $N_2$ until the end of each pyrolysis ramp and analysed by GC-MS as described in Section 3.6.1.

The pyrolytic compounds for each split were grouped into the following categories following existing literature (e.g., Carr et al., 2010; Kaal et al., 2020; Maier et al., 2025; Zhang et al., 2019; and references therein): $n$-alkanes, with $<C_{21}$ and $\geq C_{21}$ representing marine and terrestrial vegetation sources, respectively; polysaccharide derivatives (e.g., furans, furaldehydes and related compounds) from plant pigments; thiophenes derived from sulphur compounds; phenols derived from terrestrial plant lignin; polycyclic aromatic hydrocarbons (PAHs) as indicators of terrestrial carbon sources; and nitrogen (N)-containing compounds (e.g., benzonitrile, indole and related compounds) as indicators of proteinaceous aquatic microorganisms or microbial biomass. The remaining compound classes, such as cyclic alkanes and alkylbenzenes, are considered undiagnostic because they are primarily derived from recalcitrant OM (e.g., Ginnane et al., 2024; Maier et al., 2025).

### 3.7 Statistical analysis

#### 3.7.1 Soil organic matter properties

R software (version 4.2.1) was used to test whether variability in measured soil variables across five different study sites and between individual cores collected from three distinct vegetation zones within sites was statistically significant. First, Levene's test for equal variance and Shapiro-Wilk's test for normality were run on all datasets. Where the data did not meet the normality and equal variance assumptions, non-parametric Kruskal-Wallis and *post-hoc* Dunn's tests were used for pairwise comparisons. To account for the increased likelihood of Type I errors (incorrectly finding a statistically significant difference when there isn't one) associated with multiple comparisons, the Benjamini-Hochberg correction was applied to control the false discovery rate.

#### 3.7.2 Estimation of pre-restoration and post-restoration carbon accumulation rates

Changes in CARs due to saltmarsh restoration at Pāuatahanui and Robert Findlay were assessed by converting CD values into CARs for each age-depth interval of cores Pau-HM3 and Puk-MM1. The difference in OM age, and thus the degree of decay, pre- and post-restoration, makes it challenging to compare different periods. For example, an undisturbed saltmarsh will naturally have more carbon near the surface, simply due to there being less time for OM to decay; such an effect is well described for peatlands in Young et al. (2019). To compensate for this 'surface effect', we normalised, scaled and detrended the data to remove this natural increase in surface carbon and isolate the impact of restoration on carbon accumulation.



Normalisation was achieved by dividing each CAR by the maximum observed rate within each dataset, ensuring all rates are expressed relative to the maximum observed rate. Detrending involved removing the autogenic increase in near-surface carbon accumulation, which occurs in natural systems, as derived from the Awanui age-depth model, which represents the long-term

background trend of carbon accumulation expected in the absence of restoration. The mean change and percentage increase from pre- to post-restoration, from 1984 at Pāuatahanui and from 1980 at Robert Findlay, were calculated using detrended values to measure the restoration effect.

### 3.7.3 Principal Component Analysis (PCA) and hierarchical clustering

Principal Components Analysis (PCA) and hierarchical clustering were conducted on all datasets, including XRF, elemental,
stable isotope, and lipid biomarker indices. PCA has been used in recent coastal wetland studies to assess OM content and sources and explore the geochemical relationships of the various variables of interest (e.g., Carnero-Bravo et al., 2018; Fard et al., 2021; McCloskey et al., 2018; Trevathan-Tackett et al., 2023). PCA is a statistical technique commonly used to explore multivariate data for dimensionality reduction by extracting the principal orthogonal components of the data. It transforms the original variables into a new set of uncorrelated variables (principal components; PC) that capture most of the data's variability.
Visualising these components enables the identification of patterns, relationships, and dominant sources of variation within multiple datasets. PCA was performed using R Studio (version 4.2.1) with the FactoMineR package. Where the results showed low inter-variable correlation, thereby limiting the effectiveness of PCA, the variables with minimal association with PC1 and PC2 were excluded, leading to more interpretable PCs. Specifically, the quality of representation (Cos2), which identifies the contributions of the variables on each PC, and the loadings matrix, which indicates the strength and direction of the relationship
(positive or negative), were used to interpret the strength and direction of relationships.

Hierarchical clustering utilising Ward's criterion was employed to complement PCA and provide an alternative perspective on data clustering. The analysis was performed in R Studio using the hclust function. Ward's method minimises the variance within clusters, making it suitable for identifying compact groups in continuous datasets that might have interdependencies or inherent hierarchical structures. Welch's t-tests were conducted to assess the statistical significance of differences between
cluster means and the overall mean for each variable. Additionally, pairwise t-tests with the Benjamini-Hochberg correction were performed to compare the means of variables between the identified clusters.

By comparing the results of PCA and hierarchical clustering on the original datasets, we aimed to assess the robustness of the identified patterns and mitigate potential biases arising from the hierarchical sampling structure (i.e., collecting samples at multiple depths from the same core, which may have introduced sample inter-dependency in PCA). All datasets were
standardised prior to PCA and hierarchical clustering analysis.

## 4. Results

### 4.1 Stratigraphy and sedimentology

Depth of refusal, typically indicating the base of saltmarsh sediments, ranged from a minimum thickness of 5 cm for low and high marsh cores at Robert Findlay to >95 cm for a mid-marsh core at Awanui. Marsh sediments at Pāuatahanui comprised
herbaceous peat or organic-rich silts in the top 5-15 cm, underlain by silts and clays (Fig. 2a). Shelly sands and clays, interpreted as pre-marsh sediments, were observed at refusal depths across all cores. Most marsh sediments at Robert Findlay generally comprised sandy/silty peat in the top 5 cm, underlain by shelly silts and sands in high marsh zones and shelly clay in low and mid marsh zones (Fig. 2b). At Omaia, cores consisted of a thin layer of topsoil underlain by layers of intermixed silty sands, silts and clays with occasional thin peaty layers (Fig. 2c). This stratigraphy suggests the sediments have been well-
mixed due to saltmarsh drainage and conversion to pasture. Sediments at Okatakata included a 10-15 cm thick upper unit of sandy/silty peat and organic-rich silts that sit on top of silty clays (Fig. 2d). Marsh sediments at Awanui typically comprised





peaty/organic-rich silts and clays within the top 10 cm, underlain by organic-rich clays and silt (Fig. 2e,f). However, several cores collected from low marsh areas with juvenile vegetation (Awa-LM1 and Awa-LM2) were entirely composed of organic-rich sands. Most cores collected from low marsh areas at all sites were bioturbated. Examples of detailed stratigraphic logs are provided for cores Pau-HM3, Puk-MM1, Oma-MM3, Oka-MM1, and Awa-MM2 in Figures S1-S5 (Supplementary Materials).

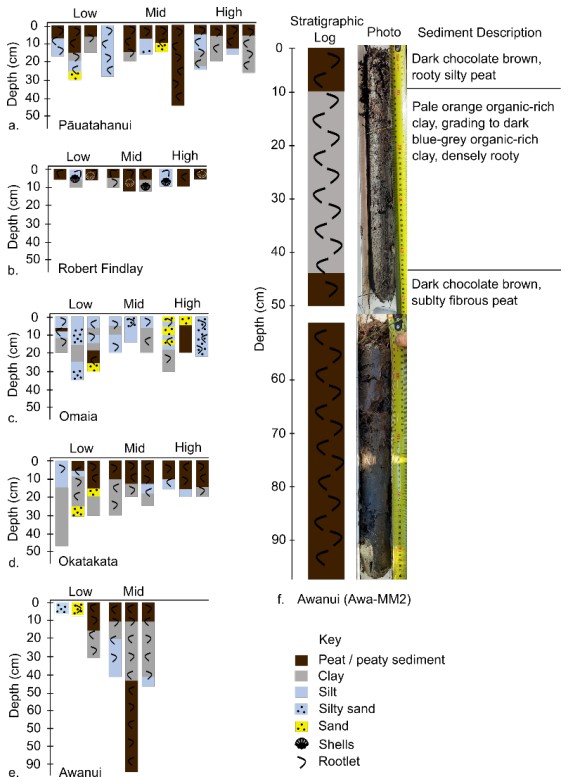

**Figure 2: Simplified stratigraphy for sediment cores from a) Pāuatahanui, b) Robert Findlay, c) Omaia, d) Okatakata, e) Awanui, and f) simplified stratigraphy and photograph of core Awa-MM2.**

## 4.2 Bulk soil organic matter variables and carbon stocks

TOC (mean ± SE, wt%) at Pāuatahanui, Robert Findlay, Omaia, Okatakata, and Awanui was 9.6 ± 0.7%, 10.0 ± 1.0%, 2.7 ± 0.4%, 5.5 ± 0.9% and 6.9 ± 0.8%, respectively (Table 1). TOC differed significantly ($p < 0.05$) between most sites, except between Pāuatahanui and Awanui, and Pāuatahanui and Robert Findlay. Mean TN (± SE, wt%) at these sites was 0.60 ± 0.04%, 0.80 ± 0.07%, 0.24 ± 0.04%, 0.37 ± 0.05% and 0.30 ± 0.03%, respectively, with significant differences between all sites except Awanui and Okatakata. Mean DBD (± SE, g cm$^{-3}$) was 0.44 ± 0.02 g cm$^{-3}$, 0.58 ± 0.06 g cm$^{-3}$, 0.64 ± 0.04 g cm$^{-3}$, 0.57 ± 0.03 g cm$^{-3}$ and 0.55 ± 0.04 g cm$^{-3}$, respectively, with significant differences observed between Okatakata and Pāuatahanui, and Omaia and Pāuatahanui. Mean CD (± SE, g cm$^{-3}$) was 0.04 ± 0.004 g cm$^{-3}$, 0.05 ± 0.01 g cm$^{-3}$, 0.02 ± 0.002 g cm$^{-3}$, 0.02 ± 0.002 g cm$^{-3}$ and 0.05 ± 0.01 g cm$^{-3}$, respectively, with significant differences between most sites except Awanui and Pāuatahanui, and Okatakata and Omaia. Significant differences in TOC, TN, CD and DBD were observed in at least one marsh zone at each site, though no consistent patterns emerged regarding which zones had higher values.

Statistical results for Levene's, Shapiro-Wilk's and Kruskal-Wallis tests are presented in Table S2. Tables S3-S7 provide the results of the *post-hoc* Dunn tests (Supplementary Materials).

**Table 1: Mean (± SE) values for TOC (wt%), TN (wt%), DBD (g cm$^{-3}$), CD (g cm$^{-3}$), carbon stocks (Mg C ha$^{-1}$) and accumulation rates (Mg C ha$^{-1}$ yr$^{-1}$ ± mean 95% confidence range) for all study sites.**



| Study Site | TOC (wt%) | TN (wt%) | DBD (g cm⁻³) | CD (g cm⁻³) | Carbon Stock (Site Mean; Mg C ha⁻¹) | Accumulation Rate (Core; Mg C ha⁻¹ yr⁻¹) |
|---|---|---|---|---|---|---|
| Pāuatahanui | 9.6 ± 0.7 | 0.60 ± 0.04 | 0.44 ± 0.02 | 0.04 ± 0.004 | 75.9 ± 16.4 | 0.98 ± 0.10 (Pau-HM3) |
| Robert Findlay | 10.0 ± 1.0 | 0.80 ± 0.07 | 0.58 ± 0.06 | 0.05 ± 0.01 | 40.7 ± 9.1 | 1.5 ± 0.76 (Puk-MM1) |
| Omaia | 2.7 ± 0.4 | 0.24 ± 0.04 | 0.64 ± 0.04 | 0.02 ± 0.002 | 52.3 ± 13.6 | - |
| Okatakata | 5.5 ± 0.9 | 0.37 ± 0.05 | 0.57 ± 0.03 | 0.02 ± 0.002 | 51.8 ± 9.3 | 0.56 ± 0.23 (Oka-MM1) |
| Awanui | 6.9 ± 0.8 | 0.30 ± 0.03 | 0.55 ± 0.04 | 0.05 ± 0.01 | 112.0 ± 100.3 | 2.5 ± 0.44 (Awa-MM2) |

355

Mean carbon stocks ranged from 40.7 ± 9.1 Mg C ha⁻¹ at Robert Findlay to 112 ± 100.3 Mg C ha⁻¹ at Awanui (Table 1; Fig. 3). The mean carbon stock was highest at Awanui (112 ± 100.3 Mg C ha⁻¹), followed by Pāuatahanui (75.9 ± 16.4 Mg C ha⁻¹), Omaia (52.3 ± 13.6 Mg C ha⁻¹), Okatakata (51.8 ± 9.3 Mg C ha⁻¹), and Robert Findlay (40.7 ± 9.1 Mg C ha⁻¹). Median distribution of carbon stocks at four of five sites shows a positive skew, which reflects the influence of sampling sites with

360   higher carbon stocks. A similar trend is observed for low and mid marsh zones at Awanui, mid marsh zone at Pāuatahanui, low and high marsh zones at Omaia, low marsh zone at Okatakata, and low and mid marsh zones at Robert Findlay. Table S8 (Supplementary Materials) summarises the results by marsh zone for each site. Kruskal-Wallis results indicate no significant difference in carbon stocks across the five sites (N=45; p=0.18).



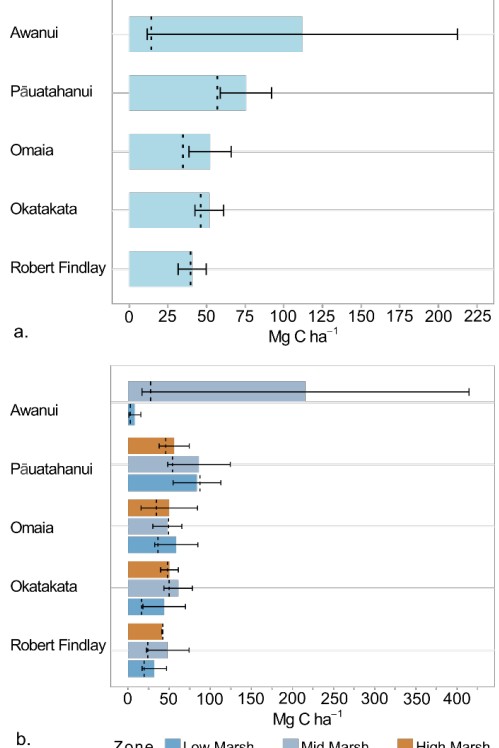

**Figure 3: a) Mean carbon stocks (Mg C ha⁻¹ ± SE) at each study site, and b) mean carbon stocks per marsh zone for each study site. The black dotted line represents the median value.**

### 4.3 Chronology

Total $^{210}$Pb activities for all three age-depth models decline from the surface and reach supported levels (i.e., equilibrium with parent isotope; $\leq$10 $^{210}$Pb Bq kg⁻¹) at 40 cm for Awanui and 20 cm at Okatakata but do not reach supported levels at Robert Findlay. $^{137}$Cs peaks at Robert Findlay, Okatakata and Awanui were evident between 9-10 cm, 2-3 cm, and 7-8 cm, respectively, (Fig. 4b,d,f) and correlate to the ~1965 peak fallout in NZ (Goff & Chagué-Goff, 1999; King et al., 2024). The mean 95% confidence range is 26.2 years at Robert Findlay, 51.2 years at Okatakata and 24.9 years at Awanui. Robert Findlay shows age uncertainty <30 years for the top 10 cm of the core, which increases up to 47 years at the base of the marsh deposit (Fig. 4a). Age uncertainty at Okatakata is below 30 years for sediments deposited down to 5 cm and increases to 55 years at the marsh base (Fig. 4c). Age uncertainty in the Awanui core for sediments deposited between 0 and 20 cm is less than 20 years. This uncertainty increases to between 20-32 years below 20 cm (Fig. 4e).





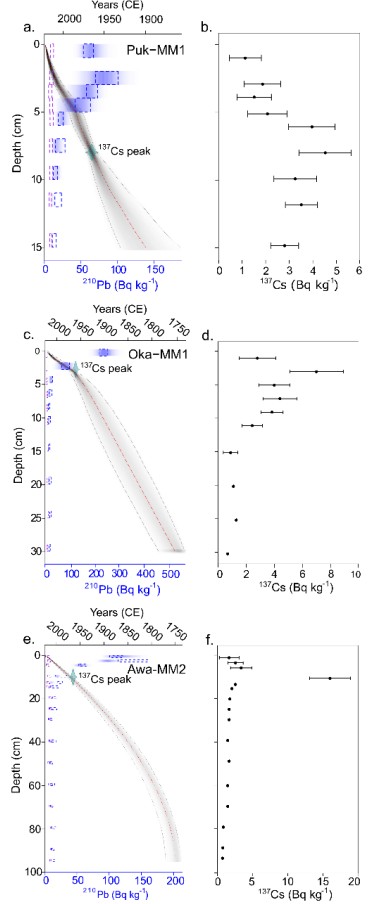

**Figure 4**: **a) Age-depth model for Robert Findlay (core Puk-MM1); c) Age-depth model for Okatakata (core Oka-MM1); and e) Age-depth model for Awanui (core Awa-MM2). The y-axis represents the depth of the cores (cm), and the x-axis depicts the estimated age (years in Common Era; CE). The model mean is represented by the red dotted line, and the grey shaded area represents the 95% confidence interval. The dotted boxes and shading represent total $^{210}$Pb (Bq kg$^{-1}$) activity. $^{137}$Cs peak is plotted as a calendar date (1965 ± 2). b), d) and f) show $^{137}$Cs levels for each core.**

### 4.4 Carbon Accumulation Rates (CARs)

Awanui exhibited the highest mean CAR of 2.5 ± 0.44 Mg C ha$^{-1}$ yr$^{-1}$ (Table 1; Fig. 5). Robert Findlay demonstrated the second-highest mean CAR of 1.5 ± 0.76 Mg C ha$^{-1}$ yr$^{-1}$. This was followed by Pāuatahanui with a rate of 0.98 ± 0.10 Mg C ha$^{-1}$ yr$^{-1}$, and Okatakata exhibited the smallest CAR of 0.56 ± 0.23 Mg C ha$^{-1}$ yr$^{-1}$.

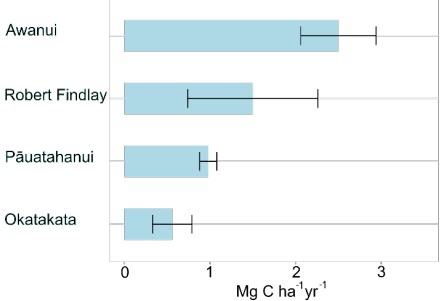

**Figure 5: Individual core CARs (Mg C ha$^{-1}$ yr$^{-1}$ ± mean 95% confidence range) dated at each study site. Pāuatahanui core Pau-HM3 was interpolated based on the age-depth model in King et al. (2024).**



**4.5 Estimated pre-restoration and post-restoration carbon accumulation rates – Pāuatahanui and Robert Findlay**

The detrended CARs showed a clear transition from negative values before restoration to positive values after restoration (Fig.

6). Negative values indicate periods where carbon accumulation was lower than expected based on the background trend, reflecting the impact of historical degradation, while positive values indicate improved carbon accumulation relative to the baseline, suggesting an increase due to restoration. The mean change in detrended accumulation rate following restoration was 0.11 and 0.32 (unitless) at Pāuatahanui and Robert Findlay, respectively. The percent increase was calculated to be 173% at Pāuatahanui and 112% at Robert Findlay, relative to the degraded state before restoration.

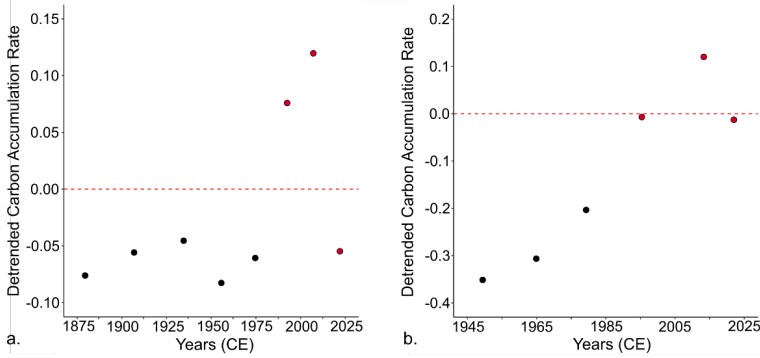


**Figure 6: Detrended CARs (unitless) at a) Pāuatahanui core Pau-HM3 and b) Robert Findlay core Puk-MM1. Black circles represent pre-restoration rates, and red circles represent post-restoration rates.**

**4.6 Stable isotope analysis of carbon and nitrogen**

$\delta^{13}C_{org}$, $\delta^{15}N$ and C:N ranges for the study sites are provided in Table 2. $\delta^{13}C_{org}$ versus C:N scatter plots (Fig. 7) show that the

sample distributions across all sites fall predominantly within the C3 plant range, freshwater dissolved OC (DOC) and particulate OC (POC) sources. Omaia and Okatakata show contributions from two additional sources of OC: freshwater algae and marine DOC. Omaia exhibits the widest range of carbon sources.





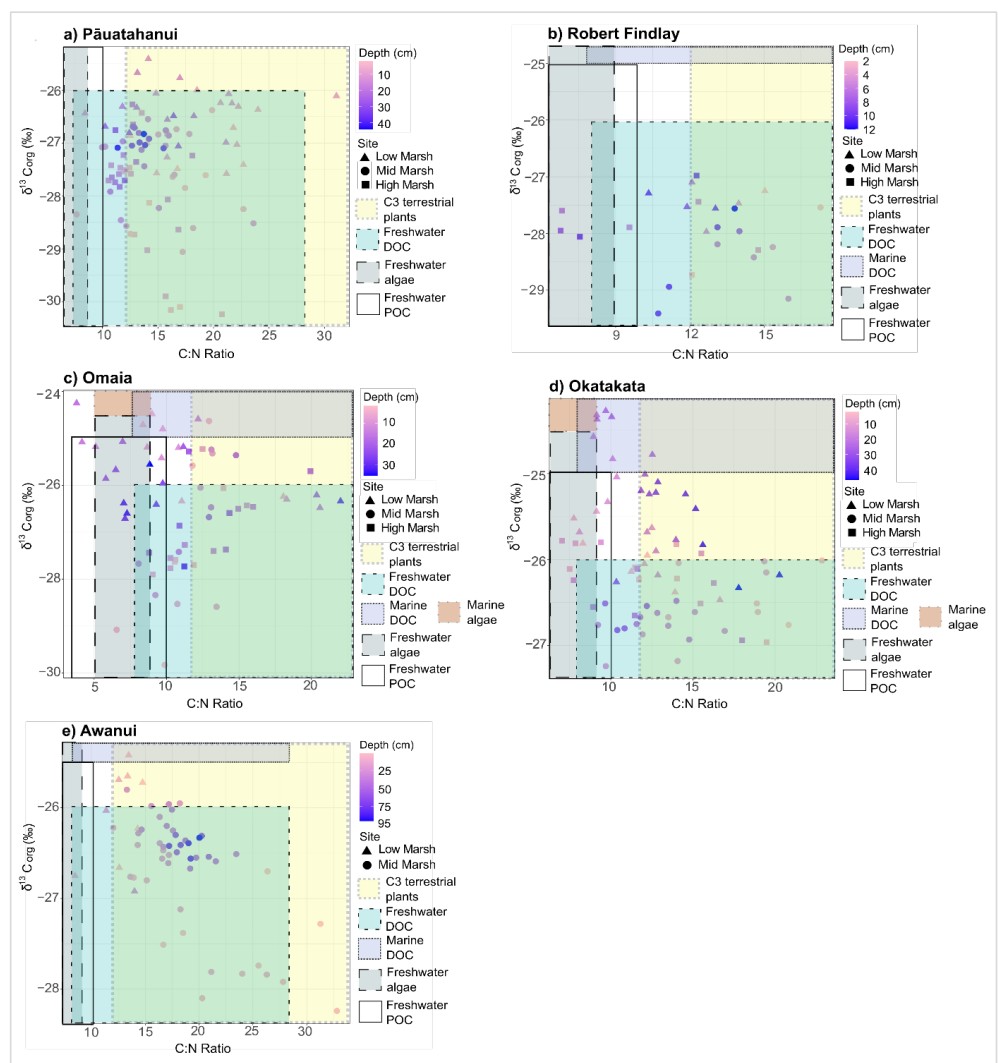

**Figure 7: Scatterplots of δ¹³C_org (‰) and C:N ratios for a) Pāuatahanui, b) Robert Findlay, c) Omaia, d) Okatakata and e) Awanui.** Note that the x and y scales differ based on site-specific ranges. The coloured boxes represent the typical δ¹³C_org and C:N ranges of the different sources of organic inputs to the coastal environment – C3 terrestrial plants, freshwater DOC, marine DOC, freshwater algae, marine algae, and freshwater POC. The ranges have been compiled from various studies and presented by Lamb et al. (2006).

**Table 2: δ¹³C_org, δ¹⁵N and C:N ratio ranges at the five study sites.**

| Site | δ¹³C_org (‰) | δ¹⁵N (‰) | C:N Ratio |
|---|---|---|---|
| Pāuatahanui | -30.2 – -25.4 | 1.9 – 8.4 | 8 – 31 |
| Robert Findlay | -29.4 – -26.6 | 3.8 – 7.2 | 7 – 17 |
| Omaia | -29.8 – -24.3 | 2.1 – 6.9 | 4 – 22 |
| Okatakata | -27.2 – -24.3 | 0.5 – 8.6 | 7 – 23 |
| Awanui | -28.2 – -25.4 | 0.9 – 7.0 | 8 – 33 |



**4.7 X-ray fluorescence (XRF)**

XRF analyses revealed the presence of the following major and trace elements in all cores: Mn, Al, Fe, Zr, Ca, Si, K, Sr, Nb, Rb, Ti, Zn, S, Pb and Cu. The elements and elemental ratios associated with OM sources and preservation were chosen for

data analysis: Ca, Sr and S as marine sources (e.g., carbonates and sulfates) due to their higher concentrations in seawater compared to freshwater, with Ca and Sr also being indicators of biogenic carbonate; S as indicative of reducing conditions/anoxic environments that result from intrusion of sulphate-rich marine waters into organic sediments where sulphate-reducing bacteria oxidise the OM; Ti, Al, Fe, Si, and K as terrestrial/lithogenic sources as these elements are generally derived from weathering of continental silicate rocks; Fe and Mn as indicative of redox processes, for example, sulphate-

reduction intensifying biogeochemical cycling of metals; and Zr:Rb as a grain size proxy (coarse-clay ratio), as Zr is predominantly found in coarser sediments and Rb in clays (Croudace & Rothwell, 2015; Ewers Lewis et al., 2019; Kelleway et al., 2017; Naeher et al., 2013). PCA and hierarchical cluster analysis were used to examine the stratigraphy and relationships between the elemental and other datasets. These results are shown in Figures 10 and 11 and discussed in Sections 5.2 and 5.3.

**4.8 Lipid biomarkers**

**4.8.1 Distribution of biomarkers, ratios and indices**

The distribution of $n$-alkanes in the apolar biomarker fractions ranges from $C_{18}$ to $C_{33}$ and shows the dominance of either mid-chain (mid-molecular weight; $C_{21}$-$C_{25}$) or long-chain (high molecular weight; $C_{26}$-$C_{33}$) $n$-alkanes across all sites, and there is a higher relative abundance of odd-carbon $n$-alkanes than even-carbon $n$-alkanes (Fig. 8). Pāuatahanui has higher relative abundances of long-chain $n$-alkanes (65%) than mid-chain $n$-alkanes (33%). In comparison, at Okatakata and Awanui, the

mid-chain $n$-alkanes (50% and 75%, respectively) exhibit a higher relative abundance than the long-chain $n$-alkanes (49.5% and 25%, respectively). Short-chain $n$-alkanes (<$C_{21}$) contributions were 2% at Pāuatahanui and <1% at Okatakata and Awanui.

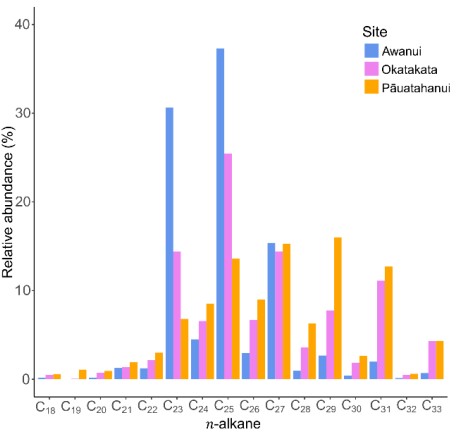

**Figure 8: Relative abundance plot of $n$-alkanes at Awanui, Okatakata and Pāuatahanui.**


The total concentrations of $C_{21}$-$C_{33}$ $n$-alkanes in sediments ranged from 0.01 to 2.9 µg g$^{-1}$ TOC at Pāuatahanui, 0.01 to 4.4 µg g$^{-1}$ TOC at Okatakata, and from 0.7 to 9.4 µg g$^{-1}$ TOC at Awanui.

CPI provides an estimate for the predominance of odd-numbered over even-numbered carbon chains and was calculated for $C_{23}$-$C_{33}$ carbon homologues using Equation 2 as reported in Wang et al. (2003):

$$CPI = \frac{\sum oddCn}{\sum evenCn}$$    (2)



CPI has been commonly used as a proxy to estimate the degree of OM degradation or determine dominant OM sources. Odd-numbered $n$-alkane chains dominate in fresh biomass and recent sediments, and diagenetic alteration of OM results in preferential decay of odd chain $n$-alkanes (Bray & Evans, 1961; Meyers & Ishiwatari, 1993). High CPI values can, therefore, indicate fresh OM, while values <1 indicate a high degree of microbial degradation or thermal maturation (Cranwell, 1981;
Eglinton & Hamilton, 1967). Values close to 1-2 may indicate some extent of OM degradation by microorganisms (Jaffé et al., 2001; Tanner et al., 2010; Zhao et al., 2024). CPI values >1 are typically interpreted to represent contributions of some types of terrestrial vascular plants, with emergent and submerged/floating aquatic plants typically exhibiting CPI values >3 (Bray & Evans, 1961; Eglinton & Hamilton, 1967; Jiménez-Morillo et al., 2024).

Similar to CPI, OEP also represents the predominance of odd-numbered over even-numbered carbon chains. This index is
calculated using Equation 3 from Zech et al. (2010).

$$OEP = \frac{C_{27}+C_{29}+C_{31}+C_{33}}{C_{28}+C_{30}+C_{32}} \tag{3}$$

High OEP values >1 have been interpreted to represent fresh, undegraded OM or terrestrial plant sources, while lower OEP values are indicative of OM degradation or less terrestrial inputs (Cranwell, 1981; Eglinton & Hamilton, 1967; Wang et al., 2003; Wang et al., 2015; Zech et al., 2010).

ACL is the weighted average of carbon chain lengths for the long chain $n$-alkanes detected in the $C_{27}$-$C_{31}$ range and was calculated following Equation 4 of Poynter & Eglinton (1990):

$$ACL = \frac{\sum[(27 \times C_{27})+(29 \times C_{29})+(31 \times C_{31})]}{\sum(C_{27}+C_{29}+C_{31})} \tag{4}$$

Variations in ACL can represent changes in vegetation types and have also been partly attributed to changes in prevailing temperature and/or moisture of the surrounding environment (Derrien et al., 2017; J. G. Poynter et al., 1989; Zhou et al., 2010).
For example, the predominance of $C_{27}$ and $C_{29}$ $n$-alkanes is characteristic of rush, sedge, shrub and tree species, while $C_{31}$, as well as $C_{33}$, are more abundant in grasses and herbs (Eley et al., 2016; Zech et al., 2010).

P(aqueous), referred to as $P_{aq}$, is calculated using Equation 5 from Ficken et al. (2000):

$$P_{aq} = \frac{C_{23}+C_{25}}{C_{23}+C_{25}+C_{29}+C_{31}} \tag{5}$$

Higher $P_{aq}$ values indicate a higher proportion of submerged and emergent aquatic vascular plants (macrophytes) and wetter
conditions (Ficken et al., 2000). Low $P_{aq}$ values of ≤0.25 indicate high contributions of higher terrestrial vascular plants, whereas high $P_{aq}$ values >0.4 indicate dominant contributions of marine and freshwater macrophytes (Ficken et al., 2000; Zhao et al., 2024). Sikes et al. (2009) further differentiate between emergent macrophytes, typically ranging between 0.4 and 0.6, and submerged and floating macrophytes exhibiting values >0.6.

Finally, the stanol-sterol ratio was calculated following Equation 6 from Naafs et al. (2019):

$$stanol - sterol\ ratio = \frac{C_{28}\ stanol+C_{29}\ stanol}{C_{28}\ stanol+C_{29}\ stanol+C_{28}\ sterol+C_{29}\ sterol} \tag{6}$$

$C_{28}$ steroids, specifically campesterol and its degradation product campestanol, and $C_{29}$ steroids, specifically sitosterol and its degradation product sitostanol, are used to calculate the stanol-sterol ratios, as these were the most abundant steroids in the samples. Stigmasterol was present in some samples but was too low in abundance to include in the ratio calculations. $C_{28}$ and $C_{29}$ sterols are derived mainly from higher plants (Gaskell & Eglinton, 1976; Volkman, 1986). In anaerobic and reducing
conditions, sterols are reduced to stanols due to microbial activity. This makes the stanol-sterol ratio a good indicator of diagenetic degradation of plant OM, where higher stanol-sterol ratios indicate more degraded material (Naafs et al., 2019; Wakeham, 1989).

CPI values ranged from 1.2 at Pāuatahanui to 10.8 at Awanui. The OEP index was >2.2 at all sites, ranging from 2.2 at Pāuatahanui to 51.4 at Awanui. ACL values at all sites ranged between 27.2 and 29.9. The $P_{aq}$ index ranged from 0.2 at
Pāuatahanui to 1.0 at Awanui. The stanol-sterol ratio fell between 0 at Pāuatahanui and Okatakata and 0.18 at Pāuatahanui. Table 3 provides the ranges for each site by vegetation zone, and Figure S6 (Supplementary Materials) shows boxplots for the



biomarker indices for each core examined in the study. Example chromatograms showing the distribution of $n$-alkanes and steroids are presented in Figures S7 and S8 (Supplementary Materials).

**Table 3: Biomarker index ranges at Pāuatahanui, Okatakata and Awanui. Mean ± SE are presented in brackets.**

| Site | Marsh | CPI | OEP | ACL | $P_{aq}$ | stanol-sterol |
|---|---|---|---|---|---|---|
| Pāuatahanui | Low Marsh | 1.2-2.4 (1.7 ± 0.2) | 2.5-4.9 (3.7 ± 0.4) | 28.3-29.1 (28.6 ± 0.13) | 0.4-0.7 (0.5 ± 0.05) | 0.03-0.08 (0.04 ± 0.01) |
| | Mid Marsh | 1.2-5.5 (3.9 ± 0.7) | 2.2-11.9 (7.8 ± 1.4) | 28.3-29.4 (29.1 ± 0.16) | 0.2-0.6 (0.3 ± 0.06) | 0.06-0.18 (0.11 ± 0.02) |
| | High Marsh | 1.5-4.6 (2.8 ± 0.6) | 3.1-7.1 (4.9 ± 0.9) | 28.5-29.4 (28.7 ± 0.17) | 0.2-0.6 (0.4 ± 0.06) | 0.00-0.05 (0.02 ± 0.01) |
| Okatakata | Low Marsh | 1.3-9.3 (4.2 ± 1.1) | 2.8-43.9 (16.0 ± 6.3) | 27.2-28.2 (27.7 ± 0.17) | 0.7-1.0 (0.8 ± 0.04) | 0.00-0.08 (0.03 ± 0.01) |
| | Mid Marsh | 1.8-4.3 (3.2 ± 0.5) | 3.3-22.7 (8.9 ± 3.6) | 28.0-29.9 (28.9 ± 0.40) | 0.3-0.8 (0.6 ± 0.10) | 0.07-0.10 (0.01 ± 0.007) |
| | High Marsh | 4.5-10.3 (7.2 ± 1.2) | 7.2-13.7 (10.4 ± 1.2) | 28.3-29.9 (29.3 ± 0.30) | 0.5-0.8 (0.6 ± 0.04) | 0.08-0.10 (0.01 ± 0.004) |
| Awanui | Mid Marsh | 7.1-10.8 (8.4 ± 0.4) | 5.6-51.4 (18.8 ± 4.9) | 27.4-28.2 (27.7 ± 0.30) | 0.9-1.0 (0.9 ± 0.01) | 0.04-0.17 (0.10 ± 0.01) |

**4.9 Ramped-Pyrolysis Oxidation-Accelerator Mass Spectrometry (RPO-AMS) and Pyrolysis-Gas Chromatography-Mass Spectrometry (Py-GC-MS)**

CRAs for the collected and measured pyrolytic splits ranged from 515 ± 85 years Before Present (BP) to 2,491 ± 165 years BP at Okatakata, with measured pyrolysis temperatures ranging from 105°C to 700°C. At Awanui, CRAs ranged from 2,350 ± 160 years BP to 3,246 ± 188 years BP, with measured pyrolysis temperatures ranging from 105°C to 700°C. The isotopic mixing model results provide estimates of 67% (first collected pyrolytic split) to 0% (fifth collected pyrolytic split) and 62% to 0% of syndepositional OC within sediments at Okatakata and Awanui, respectively. Table 4 shows the RPO-AMS and isotopic mixing results, and Figure S9 (Supplementary Materials) shows the RPO thermographs with the pyrolytic splits.

Py-GC-MS analysis shows that the OC comprises predominantly terrestrial vegetation (polysaccharide derivatives, phenols), lower amounts of marine OM (<C$_{21}$ $n$-alkanes, N-compounds) and/or soil microbial biomass (N-compounds), other recalcitrant carbon sources (cyclic alkanes and alkylbenzenes) and undiagnostic compounds. Figure 9 shows the identified relative abundances of the determined compound groups in all five measured pyrolytic splits. The most common compound types in the first split are polysaccharide derivatives, thiophenes and N-compounds, making up 72% and 85% at Okatakata and Awanui, respectively. Higher temperature splits show increasing proportions of recalcitrant OC (phenols, cyclic alkanes and alkylbenzenes) and undiagnostic sources. By the third split, these contribute 59% and 48% at Okatakata and Awanui, respectively. Figure S10 (Supplementary Materials) shows example chromatograms for the temperature splits following ramped Py-GC-MS analysis.



**Table 4: RPO-AMS and isotopic mixing model ($C_{modern}$) results for Okatakata (Oka) and Awanui (Awa) samples.**

| Sample | TOC (wt%) | CRA (yr BP) | CRA error | $F_m$ | $F_m$ error | $\Delta^{14}C$ (‰) | $\Delta^{14}C$ error | Split | Pyrolysis temperature min – max (°C) | | $C_{modern}$ (%) |
|---|---|---|---|---|---|---|---|---|---|---|---|
| Oka-MM1 28-30 cm (<90 μm) | 0.9 | 515 | 85 | 0.94 | 0.010 | -70.26 | 9.93 | 2 | 105 | 280 | 66.84 |
| | | 1208 | 31 | 0.86 | 0.003 | -147.14 | 3.36 | 3 | 280 | 333 | 41.49 |
| | | 1518 | 31 | 0.83 | 0.003 | -179.42 | 3.21 | 4 | 333 | 420 | 30.85 |
| | | 1937 | 35 | 0.79 | 0.003 | -221.08 | 3.41 | 5 | 420 | 505 | 17.12 |
| | | 2491 | 165 | 0.73 | 0.015 | -273.02 | 15.02 | 6 | 505 | 700 | -0.01 |
| Awa-MM2 90-95 cm (<90 μm) | 2.7 | 2350 | 160 | 0.75 | 0.015 | -260.18 | 14.82 | 2 | 105 | 300 | 62.35 |
| | | 2572 | 75 | 0.73 | 0.007 | -280.36 | 6.77 | 3 | 300 | 385 | 46.24 |
| | | 2900 | 36 | 0.70 | 0.003 | -309.13 | 3.14 | 4 | 385 | 444 | 23.28 |
| | | 3346 | 29 | 0.66 | 0.002 | -346.47 | 2.42 | 5 | 444 | 520 | -6.52 |
| | | 3246 | 188 | 0.67 | 0.016 | -338.26 | 15.5 | 6 | 520 | 700 | 0.03 |

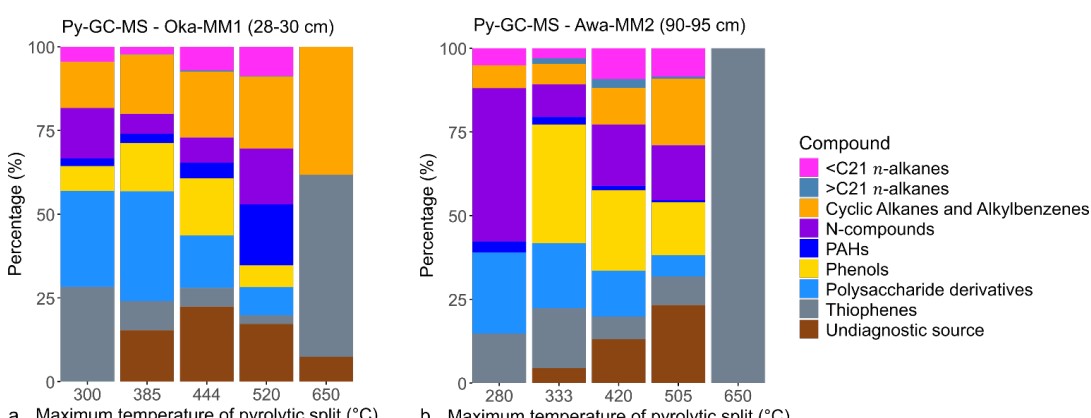

**Figure 9: Bar graphs of OC source composition for each pyrolytic split (C°) for a) Okatakata sample Oka-MM1 (28-30 cm) and b)**
**Awanui sample Awa-MM2 (90-95 cm).**

### 4.10 Principal Component Analysis (PCA) and hierarchical clustering

#### 4.10.1 Elemental, isotope and X-ray fluorescence (XRF) datasets

PCA of elemental, isotope, and XRF datasets for all samples (excluding δ15N and Mn due to their weak correlations) explained 52.9% of the variance along PC1 and PC2 (Fig. 10a). PC1 reflects OM content with positive correlations for TOC, TN, CD,
C:N, Ca, and S, and negative correlations for $\delta^{13}C_{org}$, DBD, Al, Ti, Fe, Si, K, and Zr:Rb. TOC and TN displayed the highest positive loadings and Cos2 values, suggesting they are the most influential variables in PC1. PC2 reflects lithogenic content with positive correlations for Ti, Fe, Al, DBD, S, C:N, CD, and $\delta^{13}C_{org}$, and negative correlations for TOC, TN, Si, K, Ca, Sr, and Zr:Rb. Ti and Fe had the highest positive loadings and Cos2 values, suggesting that they are the most influential variables in PC2. Hierarchical cluster analysis identified two distinct groups (Fig. 10b), with Cluster 1 (top 10 cm samples) exhibiting
statistically significantly ($p < 0.05$) higher mean TOC, TN, CD, C:N, and Ca, and lower $\delta^{13}C_{org}$, Al, Ti, Si, Fe, and Zr:Rb compared to Cluster 2 (0 – 95 cm samples). These differences highlight variations in OC content and composition with depth.



PCA of dated cores reinforced these trends, showing distinct patterns between sites (Fig. 10c). Analysis revealed four distinct clusters with statistically significant differences (Fig. 10c). Cluster 1, which includes Robert Findlay and surface Pāuatahanui samples, exhibited higher $\delta^{15}N$, Ca, Sr, and Mn, but lower $\delta^{13}C_{org}$, Ti, and Fe. Cluster 2, consisting of deeper Pāuatahanui and

Okatakata samples, showed lower TOC, TN, CD, C:N, Ca, S, and higher Al, Si, K, and Zr:Rb. Cluster 3, represented by surface Awanui and Okatakata samples, had higher TOC, TN, CD, C:N, and lower Sr, Al, Si, K, and Zr:Rb. Finally, Cluster 4, comprising deeper Awanui samples, demonstrated higher $\delta^{13}C_{org}$, S, Fe, Ti, Zr:Rb, and lower $\delta^{15}N$. These findings highlight significant geochemical variations across sampling locations and sediment depths.

**4.10.2 Elemental, isotope, X-ray fluorescence (XRF) and lipid biomarker datasets**

PCA of elemental, isotope, XRF, and lipid biomarker datasets (excluding $\delta^{15}N$, Mn, and stanol-sterol) explained 57.7% of the variance along PC1 and PC2 (Fig. 11a). PC1 represents lithogenic content with positive loadings for Ti, Fe, DBD, S, CPI, OEP, and $P_{aq}$, and negative correlations for Sr, Si, K, and ACL. PC2 is associated with OM content, showing positive correlations for TOC, TN, CD, C:N, and Ca, and negative correlations for Al, Zr:Rb, and $\delta^{13}C_{org}$. Hierarchical clustering revealed three distinct groups with statistically significant differences in means (Fig. 11b). Cluster 1 (top 15 cm from

Pāuatahanui, Okatakata, and Awanui) showed higher TOC, TN, CD, and Ca, but lower Si and Zr:Rb. Cluster 2 (15-45 cm from Pāuatahanui) had lower TOC, TN, CD, Ca, S, Fe, CPI, and OEP, but higher Sr, Si, and Zr:Rb, while Cluster 3 (5-95 cm from Okatakata and Awanui) exhibited higher $\delta^{13}C_{org}$, S, Fe, Ti, and Paq, and lower Sr and ACL.

PCA of cores with age-depth estimates from Pāuatahanui, Okatakata, and Awanui (including stanol-sterol data) highlighted site-specific variations (Fig. 11c). Awanui samples clustered positively on PC1 and PC2, indicating higher values of lithogenic

variables (S, Fe, Ti, DBD, $\delta^{13}C_{org}$, CPI, $P_{aq}$, stanol-sterol) below 20 cm and higher OM variables (TOC, TN, CD, Ca) in younger surface samples. Okatakata samples were broadly distributed, with deeper samples clustering negatively on PC2 (lithogenic variables) and surface samples positively on PC1 (OM variables). Pāuatahanui samples clustered negatively on PC1, indicating higher Si, K, and Sr. Hierarchical clustering identified three distinct clusters (Fig. 11d). Cluster 1, comprising young surface samples from Pāuatahanui, Okatakata, and Awanui, showed higher levels of OM content and lower lithogenic content. Clusters

2 and 3, representing older, deeper samples, displayed trends of decreasing OM and increasing CPI, OEP and stanol-sterol with increasing depth.



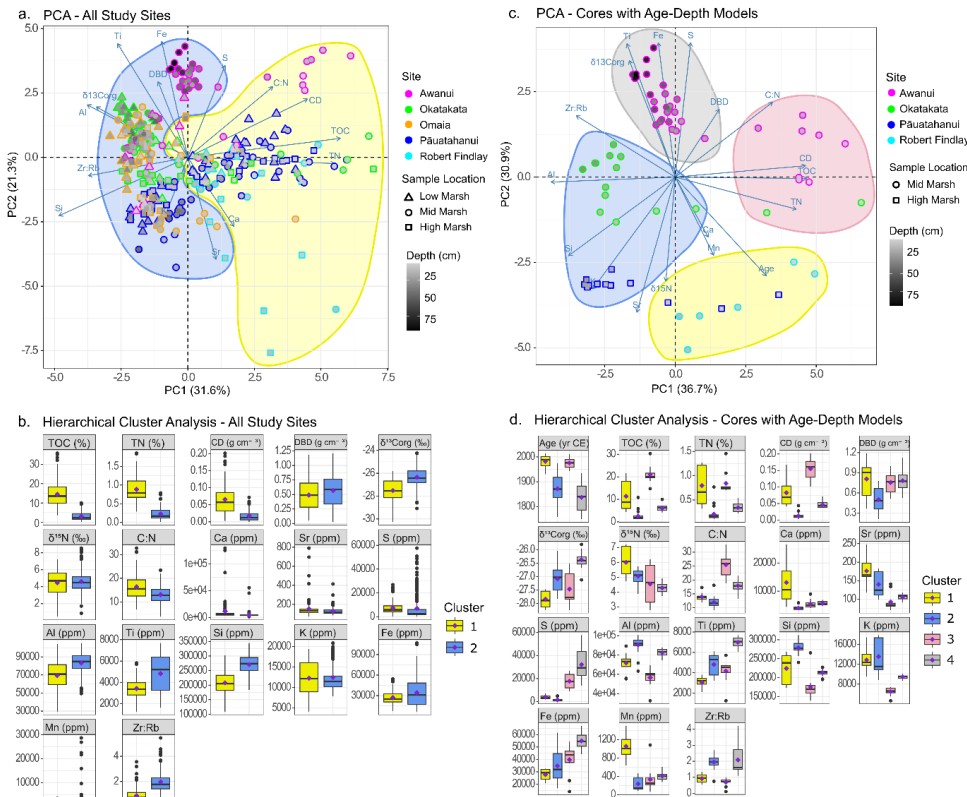

**Figure 10: a) PCA plots and b) hierarchical cluster analysis of elemental, isotopic and XRF data sets for all study sites. c) PCA plots and d) hierarchical cluster analysis of elemental, isotopic and XRF data sets for cores with age-depth models from Pāuatahanui (Pau-HM3; interpolated based on the depth-age model by King et al. (2024)), Robert Findlay (Puk-MM1), Okatakata (Oka-MM1) and Awanui (Awa-MM2). The clusters identified in hierarchical cluster analysis are outlined on the PCA plots.**




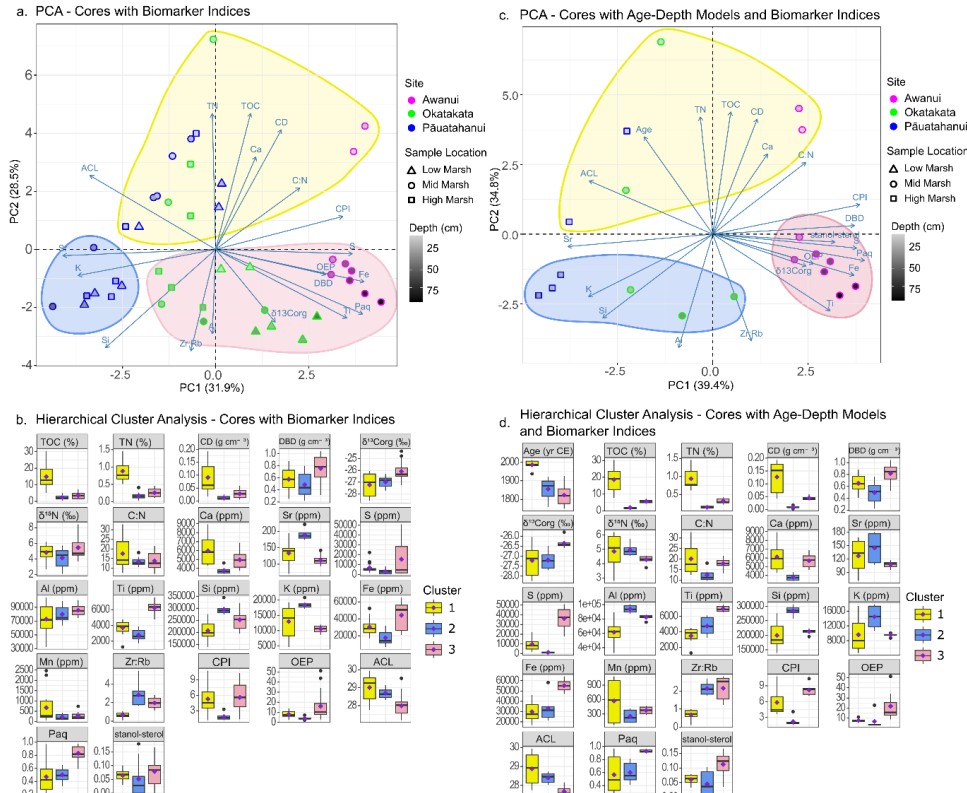

**Figure 11: a) PCA plots and b) hierarchical cluster analysis of elemental, isotopic, XRF and biomarker data sets for cores from Pāuatahanui (Pau-HM1), Okatakata (Oka-MM1) and Awanui (Awa-MM2). c) PCA plots and hierarchical cluster analysis of elemental, isotopic, XRF and biomarker data sets for cores with age-depth models from Pāuatahanui (Pau-HM1; interpolated based on the depth-age model by King et al. (2024)), Okatakata (Oka-MM1) and Awanui (Awa-MM2). The clusters identified in hierarchical cluster analysis are outlined on the PCA plot.**

## 5 Discussion

This study quantified carbon stocks, accumulation rates, OM sources and preservation across five saltmarsh locations in NZ. We found soil OC stocks and accumulation rates across the five saltmarsh sites to be highly variable, ranging from $40.7 \pm 9.1$ to $112 \pm 100.3$ Mg C ha$^{-1}$ and $0.56 \pm 0.23$ to $2.5 \pm 0.44$ Mg C ha$^{-1}$ yr$^{-1}$, respectively (Mean $\pm$ SE; Table 1). These values are similar to previously reported mean values for saltmarshes and mangroves in NZ and greater than intertidal unvegetated habitats (mudflats/sandflats; Bulmer et al., 2024). Values at Pāuatahanui, Robert Findlay and Okatakata are notably less than global saltmarsh averages (Chmura et al., 2003; Maxwell et al., 2024; Ouyang and Lee, 2014; Wang et al., 2021). CARs typically increase following saltmarsh restoration, which highlights the positive impact of remedial environmental actions on carbon storage potential (Fig. 6). Vegetation type (zonation) appears to have no influence on TOC stock (Tables S2-S7), but accumulation of OC in the top 10 cm of sedimentary packages, proximity to allochthonous inputs, and site-specific geochemistry all appear to influence site variability (Fig. 10 and 11). Results from carbon and nitrogen isotope (Fig. 7) and lipid biomarker analyses (Fig. 8) indicate substantial contributions from saltmarsh vegetation to the OC pool. OM decay increases with depth at all sites (Fig. 10 and 11). However, labile plant-derived OC being abundant in lowermost saltmarsh sediments under anoxic conditions suggests long-term preservation of OC (Fig. 9). Figure 12 provides an overview of the key findings.



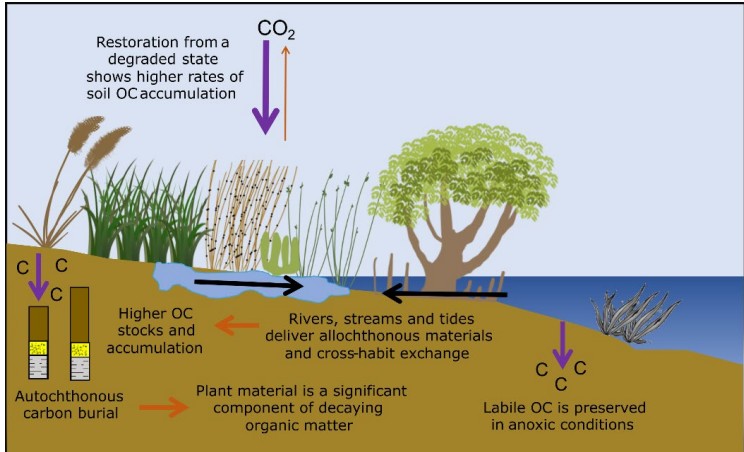

**Figure 12: Schematic diagram showing a summary of key research findings.**

**5.1 Carbon stocks and accumulation rates**

TOC stock at Pāuatahanui (75.9 ± 16.4 Mg C ha⁻¹) is comparable to estimated global representative soil OC stocks (79.2 ±

38.1 Mg C ha⁻¹) derived from analysis of soils with an average depth of 30 cm (Maxwell et al., 2024). In contrast, the mean TOC stocks at Robert Findlay (40.7 ± 9.1), Okatakata (51.8 ± 9.3), and Omaia (52.3 ± 13.6 Mg C ha⁻¹) are lower than the global average. Low values at Robert Findlay are likely because saltmarsh deposits at that location are only 9 cm thick on average. Saltmarsh sediments at Okatakata are reasonably thick (25 cm on average), and lower TOC stocks are likely due to site-specific geomorphic settings, such as lower siliciclastic inputs. Saltmarsh deposits at Omaia are 23 cm thick on average but

were subject to increased OM decay rates due to greater exposure to oxygen when the marsh was converted to pasture (Ewers Lewis et al., 2018; Heuscher et al., 2005). Therefore, these TOC stocks likely reflect a higher-than-average amount of degraded marsh materials and soils accumulated since drainage. Awanui recorded the highest mean TOC stock (112 ± 110.3 Mg C ha⁻¹) and showed the biggest range in values due to cores collected from juvenile to well-established marsh. It also contains the thickest sequence of saltmarsh sediments from one sampling site (95 cm). This thick deposit has a TOC stock value (613 Mg

C ha⁻¹) that is six-fold higher than the national average for 1 m-thick saltmarsh deposits (92.5 ± 12.42 Mg C ha⁻¹; Bulmer et al., 2024), and three-fold higher than the estimated global representative soil OC stock (231 ± 134 Mg C ha⁻¹) calculated for deposits with a similar average thickness (Maxwell et al., 2024).

Okatakata and Awanui saltmarshes are less than 700 m apart, but the mean TOC stock at Awanui is more than double that at Okatakata. Each marsh formed at a similar time (1750 ± 32 CE at Awanui and 1759 ± 54 CE at Okatakata), but the mean

sediment accumulation rate is 1.1 mm yr⁻¹ at Okatakata and 3.5 mm yr⁻¹ at Awanui. Awanui saltmarsh is closer to the Awanui River than the Okatakata saltmarsh and likely receives a relatively high input of allochthonous sediment flowing in the river. Okatakata is located further away from the river's path and likely receives less allochthonous material. These observations and our inferred cause of the variation in TOC stocks at closely located saltmarshes are not unprecedented. Previous studies have attributed higher TOC stocks in saltmarshes that are adjacent to freshwater sources to the addition of terrestrial sediments

(Hansen et al., 2017; Hayes et al., 2017; Kelleway et al., 2016; Peck et al., 2025; Van de Broek et al., 2016; Van De Broek et al., 2018). The higher rate of mineral sediment deposition leads to a more rapid advection of the deposited OC to deeper soil layers, enhancing its preservation (Kirwan & Mudd, 2012; Van de Broek et al., 2016, 2018), contributing to the higher stocks. Mean CARs at our study sites vary between 0.56 ± 0.23 and 2.5 ± 0.44 Mg C ha yr⁻¹. These rates, except for Awanui, are lower than global average values between 1.67 to 2.45 Mg C ha yr⁻¹ (Chmura et al., 2003; Ouyang & Lee, 2014; Wang et al.,

2021) but are close to the national mean estimate of 0.89 Mg C ha yr⁻¹ (Bulmer et al., 2024). Several factors likely contribute to these low CARs and include age, climate, accommodation space, and biological influences. Young saltmarsh deposits at



locations in Great Britain (Smeaton et al., 2024) and Scandinavia (Leiva-Dueñas et al., 2024) have accumulation rates (mean of $1.1 \pm 0.43$ Mg C ha yr⁻¹ and a median rate of 0.32 Mg C ha yr⁻¹, respectively) that are comparable to sites examined in this study. These rates are also significantly lower than global mean estimates and are attributed to the relatively young age of these

marshes and the fact that relatively thin saltmarsh deposits form under temperate climatic conditions. Saltmarsh deposits in the southwest Atlantic (Martinetto et al., 2023) also record a low mean carbon burial rate (0.48 Mg C ha⁻¹ yr⁻¹). Lower CARs at this location are attributed to the influence of biological activity, including bioturbation by burrowing crabs that can mix and oxygenate the soils, thereby degrading OC. Coastal sediments require accommodation space in which to accumulate. Accommodation space is created as sea levels rise and the vertical space available for mineral and organic material

accumulation expands (Rogers et al., 2019, 2022). An increase in relative sea level at the coast can occur when climate warms and causes an increase in ocean mass, as ice sheets and glaciers melt, or when land subsides due to processes including tectonics and sediment compaction (Gregory et al., 2019). Sea levels in much of the Southern Hemisphere have remained relatively stable over the past 6 kyr (Rogers et al., 2023), and saltmarsh deposits in NZ seldom exceed 0.5 meters in thickness (Gehrels et al., 2008), which partly reflects these stable sea levels. Saltmarsh deposits studied here are young, bioturbated, and

accumulated at locations where relative sea level has remained relatively stable. These factors all contribute to the lower stocks and CARs observed in this study.

### 5.1.1 Impact of saltmarsh restoration

Results from Pāuatahanui and Robert Findlay saltmarshes highlight the impact of restoration efforts on CARs and show that significantly higher rates occur following restoration (Fig. 6). This restoration effect has also been observed in saltmarshes

across the United Kingdom and northwest Europe (Mason et al., 2022). CARs at restored saltmarsh sites in these regions increased by a factor of approximately 1.6, likely due to a rapid increase in organic and mineral sediment accumulation following restoration, which is expected to slow over time (Drexler et al., 2020; Miller et al., 2022; Mossman et al., 2022). This trend is observed at the study sites, with rates increasing significantly following restoration and slowing down in the last decade.

### 5.2 Sources of organic matter in saltmarshes

Knowing the source of OM in saltmarsh soils can help us understand the processes that drive carbon accumulation and can better inform the carbon mitigation potential of these ecosystems. $\delta^{13}C_{org}$ and C:N data indicate that carbon that has accumulated in saltmarsh soils examined in this study is primarily derived from terrestrial C3 plants and estuarine biota (Fig. 7). Sediments at Okatakata, Omaia and Awanui also include OM from freshwater algae that was most likely transported to the

sites via plumes from the Awanui River. Omaia sediments contain carbon that is sourced from the widest range of sources. This feature most likely reflects anthropogenic influence on the site, including drainage and farming, with associated activities that thoroughly mixed older marine sediments with saltmarsh deposits and younger topsoil. Since both autochthonous and allochthonous sources contribute to carbon accumulation, the estimated CARs presented in this study do not solely reflect the amount of OC directly sequestered from the atmosphere through in-situ production (Smeaton et al., 2024). However, the burial

and preservation of allochthonous OM, particularly if not mineralised upon deposition, can still play a significant role in the long-term carbon storage capacity of saltmarshes (Houston et al., 2024; Smeaton et al., 2024; Van De Broek et al., 2018). Given the overlap in the isotopic and elemental ratios, it is challenging to quantify the relative contribution of each source without knowing the $\delta^{13}C_{org}$, $\delta^{15}N$ and C:N specific values for each end-member and their decomposition rates (Kumar et al., 2020a; Wang et al., 2003). Smeaton et al. (2024) quantified the in-situ contribution of saltmarsh plant carbon to the sediment

OM in the marsh systems across Great Britain, obtaining contributions ranging from 10% to 99%. This wide range reflects the dynamic settings in the coastal environment, where marine and freshwater sources and cross-habitat exchange of carbon influence OM production and accumulation (Alongi, 1997; Bulmer et al., 2020; Pondell & Canuel, 2022). Overall, the results



of stable isotope analysis indicate that C3 marsh plant inputs contribute significantly to the in-situ OM production across the studied sites. However, in coastal settings, tidal mixing and river discharge deliver allochthonous marine- and terrigenous-
derived OM, overlapping the isotopic values and reflecting a mixture of OM from different sources.

Lipid biomarker analysis offers additional insights into the origins of OM (e.g., Naeher et al. 2022, Peters et al., 2005). The dominant occurrence of $n$-alkanes in the range $C_{21}$ to $C_{33}$ (Pāuatahanui = 98%, Okatakata = 99% and Awanui = 99%), with the predominance of odd-chain carbon numbers, is attributed to epicuticular waxes from plants (Fig. 8). The contributions from marine and freshwater algae, bacteria and/or other microorganisms ($<C_{20}$) were insignificant across all sites (Cranwell,
1981; Eglinton & Hamilton, 1967; Ficken et al., 2000; Kumar et al., 2019; Meyers, 1997; Zhang et al., 2024). Additionally, positive correlations between CPI and OEP with values >3 in surface sediments suggest that saltmarsh vegetation significantly contributes to the sedimentary OM, which is characterised by relatively high stability (Fig. 10; Kennicutt et al., 1987; Zhao et al., 2024).

Specifically, the dominance of $C_{23}$ and $C_{25}$ $n$-alkanes and $P_{aq}$ values >0.6 at Okatakata and Awanui point to the influence of
saltmarsh succulent species such as *Salicornia quinqueflora* (Tanner et al., 2007, 2010). Other aquatic plants that could be contributing to the mid-chain $n$-alkanes include moss species *Kindbergia* spp. and *Polytrichum* spp. observed during sampling in the low and mid marsh areas (Ortiz et al., 2016). It is also possible that floating macrophytes are delivered via tidal/riverine inputs; however, these were not observed during data collection. In contrast, dominant peaks at $C_{27}$, $C_{29}$, $C_{31}$ and $C_{33}$ and $P_{aq}$ <0.6 at Pāuatahanui reflect emergent macrophytes and salt-tolerant terrestrial plants. Studies have demonstrated that C3
saltmarsh grass, rush, sedge and shrub species exhibit maximum abundances in the $C_{27}$-$C_{33}$ range, consistent with the $n$-alkane distributions, as well as ACL values of 27-29 (Eley et al., 2016; Ferreira et al., 2009; Ortiz et al., 2011; Pondell & Canuel, 2022; Tanner et al., 2007, 2010; Wang et al., 2003; Zhang & Wang, 2019). PCA plots confirm these observations as ACL and $P_{aq}$ display opposite trends. When $P_{aq}$ increases, indicating the dominance of non-emergent macrophytes, ACL decreases (Fig. 11). Upstream terrestrial plants, such as *Pinus radiata*, may provide additional sources of $C_{29}$ $n$-alkane (Gonzalez-Vila et al.,
2003; Kumar et al. 2020b). However, there are no adjacent pine plantations surrounding the study sites, and no terrestrial plant debris/litter was observed during data collection. Other studies have also shown that seagrass species contribute $C_{29}$ $n$-alkanes to the OC pool (Jaffé et al., 2001; Kumar et al., 2020b). Seagrass habitats are present at Pāuatahanui (Zabarte-Maeztu et al., 2020) and Rangaunu Harbour has one of the largest seagrass habitats in the country (Bulmer et al., 2024), where cross-habitat exchanges could be contributing to the composition of the OC pool (Bulmer et al., 2020).
In summary, even though lipid biomarkers typically account for 1-10% of the total OM and may not represent the gross features of organic inputs, the compound-specific information provides insight into the dominant OM sources (Meyers, 1997; Naeher et al., 2022; Zhao et al., 2024). Lipid biomarker analysis, therefore, demonstrates that saltmarsh vegetation makes up a substantial portion of OM at the studied sites. However, it is important to note that plant-derived OC tends to be more resistant to microbial breakdown than OC from algae and bacteria, which may result in its preferential preservation in sediments
(Schmidt et al., 2011; Zhang et al., 2024).

## 5.3 Organic matter preservation

The strong positive correlation between TOC and TN at all sites suggests that they are likely influenced by similar geochemical processes (Fig 10 and 11; Brandini et al., 2022; Zhang et al., 2024). Surface samples generally exhibit higher mean OM contents (TOC, TN, CD, C:N) and low mean concentrations of lithogenic components (Si, K, Al, Ti, Fe, Zr:Rb). This indicates
that young, finer-grained, organic-rich surface sediments with fresh OC are predominant across study sites (Kelleway et al., 2017; Krüger et al., 2024; Mazarrasa et al., 2023). Finer-grained samples typically have higher OC content because a higher proportion of silt and clay in the sediments enhances the preservation of OM (Mazarrasa et al., 2023; Meyers, 1994; Russell et al., 2023). In contrast, older, deeper, coarser-grained samples generally show lower OM, higher lithogenic content, enriched $\delta^{13}C_{org}$, higher stanol-sterol values, lower C:N ratios, and lower CPI values of 1-2. These characteristics suggest that OM has

been preferentially utilised and decomposed by microbial activity over time, increasing the lithogenic contribution in the OC
pool (Benner et al., 1987; Jaffé et al., 2001; Krüger et al., 2024; Zhao et al., 2024). Omaia shows enriched $\delta^{13}C_{org}$, low TOC,
TN, CD, and increased DBD values for most of the samples, reflecting the degraded and compacted nature of the soils.

At Awanui, deeper samples display higher OM content alongside S, suggesting the preservation of OC under sulphur-rich
anoxic conditions (Antler et al., 2019; Froelich et al., 1979; Thamdrup et al., 1994). This aligns with the observation that oxic
and suboxic degradation of OC is restricted to shallow depths (<20 cm), and deeper soil horizons remain more consistently
anaerobic, which promotes preservation (Howarth & Teal, 1979; Spivak et al., 2019). However, whether aerobic or anaerobic
conditions prevail also depends on the position of the saltmarsh in the tidal frame, with locations less regularly inundated
typically exhibiting more oxygenated conditions even to depths of 30 cm (e.g., Mueller et al., 2019). In these locations, other
factors controlling preservation of OM, such as microbial community compositions, formation of organo-mineral complexes
and physical protection by mineral aggregates, may be more influential (Barber et al., 2017; Macreadie et al., 2025; Spivak et
al., 2019). Deeper samples at Awanui and Okatakata also have the highest Ti, Al and Fe contents. According to the age-depth
models, the greater contribution of allochthonous-derived material for these samples encompasses the period of flood control
and drainage works carried out in the Awanui River catchment between 1916 and 1936, which likely resulted in large amounts
of terrestrial materials transported down the river to the harbour (Cathcart, 2005). Therefore, it is possible that the deposition
of mineral sediments and their geochemical interactions with OC (e.g., OC binding with iron oxides to form stable Fe-OC
complexes) contributed to stronger preservation (Barber et al., 2017; Macreadie et al., 2025). The degree of preservation
appears to be higher under anoxic conditions (Fig. 10), consistent with observations of stronger organo-mineral bonds in anoxic
submerged environments (Liu & Lee, 2006; Macreadie et al., 2025).

The increased $\delta^{15}N$ values and Mn concentrations in soil samples at Pāuatahanui and Robert Findlay can be attributed to
increased OM decay rates, possibly due to elevated nutrient inputs (e.g., nitrogen and manganese fertilisers). The sites were
farmed until recently, and adjacent areas around both reserves continue to be farmed today. Enhanced primary production
because of increased nutrient loading leads to higher rates of OM decomposition and removal of nitrogen via denitrification
and nitrification processes, enriching the residual N soil pool in $\delta^{15}N$ (Amundson et al., 2003; Peng et al., 2016; Thomson et
al., 2025; Watson et al., 2022). Pāuatahanui and Robert Findlay also have the highest mean TOC and TN in surface soils,
which could potentially be the result of larger nutrient loads increasing primary productivity of the ecosystem, at least in the
short term, as has been shown at other eutrophic coastal wetlands (Asanopoulos et al., 2021; Cuellar-Martinez et al., 2020;
Geoghegan et al., 2018; Sanders et al., 2014). In the long term, nutrient enrichment can weaken the soil structure and reduce
OM stability, thereby enhancing microbial respiration of the stored carbon (Cuellar-Martinez et al., 2020; Geoghegan et al.,
2018; Macreadie et al., 2025; Thomson et al., 2025). Pāuatahanui and Robert Findlay also have higher marine biogenic
components (Ca, Sr). This is consistent with field observations where marsh plants had colonised shelly substrate.

The isotopic mixing model results from RPO-AMS analysis provide a first-order approximation of the relative contributions
of labile, syn-depositional (autochthonous and allochthonous) versus older, recalcitrant OC in the soils. The results suggest a
relative proportion of two-thirds labile and one-third recalcitrant OC at Okatakata and Awanui basal samples for the lowest
temperature split (Table 4). RPO-AMS ages of basal samples at Okatakata and Awanui are between 324-2300 years and 2150-
3046 years older (CE), respectively, than those estimated using $^{210}Pb$ age-depth models, representing the mixture of labile and
recalcitrant carbon delivered to the site. Awanui has much older carbon compared to Okatakata. This is consistent with the
core's proximity to the river, which delivers allochthonous material from the upper catchment.

Py-GC-MS results provide insights into the composition of OC in the samples, corresponding directly to the RPO-AMS ages.
The lower temperature, younger splits of the basal samples from Okatakata (100 to 300°C and 300 to 385°C) and Awanui (100
to 280°C and 280 to 333°C) have higher relative contributions of syn-depositional OC. These lower-temperature splits release
more labile and volatile compounds, including those also obtained in the biomarker TLEs, which are more closely associated
with the sources and state of syn-depositional carbon (Ginnane et al., 2024; Rosenheim et al., 2008). Further heating during



pyrolysis (typically >400°C) breaks down the macromolecular structure of OM compounds, releasing pre-aged, transported and more recalcitrant OM, which requires higher temperatures during pyrolysis to be detected (Ginnane et al., 2024; Maier et al., 2025). This is evident in the Py-GC-MS results, where the third collected pyrolytic splits for Okatakata and Awanui produce higher relative abundances of cyclic alkanes/alkylbenzenes and other undiagnostic or refractory compounds found in the higher temperature fractions (Fig. 9).

Furans are interpreted as indicators of higher molecular weight polysaccharides such as cellulose. These carbohydrate compounds in plant cell walls are easily consumed by microbes (Carr et al., 2010; Kaal et al., 2020). Phenols, interpreted as indicators of lignin, represent the more recalcitrant OC pool. Lignin is a macromolecular compound found almost exclusively in the cell walls of terrestrial vascular plants and is more resistant to OM degradation (Grandy & Neff, 2008; Kaal et al., 2020; Zhang & Wang, 2019). Py-GC-MS results confirm contributions from saltmarsh vegetation to the OC pool at Okatakata and Awanui, indicating preservation of plant-derived labile and recalcitrant OC fractions under anoxic conditions, reinforced by enrichment in sulphur compounds such as thiophenes (González-Pérez et al., 2012). However, N-containing pyrolysis products (e.g., indole, benzonitrile, pyridine) in coastal wetland soils have also been attributed to amino acids and proteins from algae and phytoplankton (Carr et al., 2010; Kaal et al., 2020), and bacterial biomass within soil OM (Ferreira et al., 2009; Zhang & Wang, 2019; Zhu et al., 2016), suggesting allochthonous contributions and/or microbial decomposition of OC. TLE and Py-GC-MS analyses of the basal samples at Okatakata and Awanui indicate ≤1% and ≤5% contribution from marine source $n$-alkanes, respectively (Fig. 7 and 8), suggesting that N-compounds likely correspond to microbial biomass. Nevertheless, collectively, these results demonstrate that while some microbial decomposition may have occurred post-deposition, the dominant presence of syn-depositional OC, originating from plant biomass and preserved under anoxic conditions, implies effective long-term burial of OC in saltmarsh soils.

## 5.4 Implications and future research

This study provides the first integrated assessment of carbon stocks, accumulation rates, and OM sources and preservation across a latitudinal gradient of saltmarshes in NZ. Our findings reveal that carbon accumulation and preservation are strongly influenced by site-specific factors such as land use history, sediment characteristics, and allochthonous inputs. Notably, we demonstrate that even relatively young saltmarsh deposits can store substantial amounts of carbon when conditions favour OM preservation. Our results also corroborate previous research showing significant variability in soil OM properties, carbon stocks and accumulation rates across different geomorphic settings, land use, tidal regimes and salinity, marsh zones and vegetation types, lithology and soil depths (e.g., Ewers Lewis et al., 2019; Hansen et al., 2017; Kelleway et al., 2016; Martinetto et al., 2023; Martins et al., 2022; McMahon et al., 2023; Owers et al., 2020; Ruiz-Fernández et al., 2018; Russell et al., 2023; Saintilan et al., 2013). Sampling strategies for blue carbon thus need to consider the variability within the ecosystem for accurate assessments. Our results suggest that allochthonous inputs, particularly proximity to freshwater sources such as rivers, play a critical role in enhancing carbon accumulation and preservation. The study also highlights the importance of measuring OC to the base of the saltmarsh deposit to capture the full extent of carbon storage and preservation processes. Future research could further explore how environmental variables such as distance to tidal creeks and freshwater sources, vegetation biomass, elevation, inundation frequency and duration, salinity and sediment composition influence OM dynamics and carbon stocks in NZ saltmarshes (e.g., Hansen et al., 2017; Janousek et al., 2025; McMahon et al., 2023; Puppin et al., 2024; Russell et al., 2023). Furthermore, it is essential to quantify the autochthonous and allochthonous sources and their preservation characteristics to determine their contributions to carbon mitigation (Macreadie et al., 2025). Further data analysis, such as employing RPO-AMS (e.g., Houston et al., 2024; Van De Broek et al., 2018) coupled with Py-GC-MS (e.g., Kumar et al., 2019; 2020a; 2020b) along core profiles to trace OM sources and quantify labile versus recalcitrant fractions, will enhance our understanding of OC sources and decay rates. This, in turn, will lead to improved estimation approaches for carbon mitigation (e.g., Needelman et al., 2018) and more accurate evaluations of the long-term carbon sequestration capacity of BCEs.



**6 Conclusions**

This study successfully quantified carbon stocks and accumulation rates across five saltmarsh sites in NZ, revealing significant variability influenced by site-specific factors such as geomorphic settings, land use, and proximity to allochthonous inputs. The findings indicate that while carbon stocks, ranging from $40.7 \pm 9.1$ to $112 \pm 100.3$ Mg C ha$^{-1}$, and accumulation rates, ranging from $0.56 \pm 0.23$ to $2.5 \pm 0.44$ Mg C ha$^{-1}$ yr$^{-1}$, at the majority of the sites are lower than global averages, they are

comparable to national estimates. Notably, the study highlights the positive impact of saltmarsh restoration on carbon accumulation rates. Stable isotopes, lipid biomarkers, and RPO-MS with Py-GC-MS analyses, offer insights into the sources and composition of organic materials in the soil. The analyses reveal significant contributions from saltmarsh vegetation and highlight the preservation of plant-derived OC over centuries. The findings of this study will improve national estimates of carbon accumulation in saltmarsh ecosystems and the methods employed to assess BCEs contributions to climate mitigation.

**Data availability**

To view data for this article, please visit https://doi.org/10.5281/zenodo.15702914.

**Supplement**

The supplement related to this article is available online at:

**Author contributions**

OA: conceptualisation, funding acquisition, project administration, investigation, methodology, data curation, formal analysis, visualisation, writing (original draft preparation, review and editing); JR: conceptualisation, supervision, writing (review and editing); RL: conceptualisation, funding acquisition, supervision, writing (review and editing); SN: investigation, visualisation, writing (review and editing); DK: investigation, visualisation, writing (review); CG: investigation, writing (review and editing); JT: supervision, writing (review); MB: investigation, writing (review); CW: investigation; JD: investigation; JC:

investigation; AP: investigation.

**Competing interests**

Some authors are members of the editorial board of Biogeosciences.

**Acknowledgements**

We thank Jay Streatfield for his assistance with field work and Dr. Michael Lechermann (ESR Christchurch) for the generation

of $^{210}$Pb data. Dr Rewi Newnham (Victoria University of Wellington), Dr Gavin Dunbar (Victoria University of Wellington), Dr Kate Clark (GNS Science) and Dr Joe Prebble (GNS Science) are thanked for their valuable discussions and help with securing field gear and sample storage space. We thank Jane Chewings and Dez Tessler (Victoria University of Wellington) for their assistance with H&S planning and oversight of field and laboratory work. We thank Pūkorokoro Miranda Naturalists Trust, the Department of Conservation and the landowners of Omaia Island for allowing sampling access. We acknowledge

the study was conducted on the ancestral lands of Ngāi Takoto, Ngāti Pāoa and Ngāti Toa.



**Financial support**

This research was supported by the Ministry of Business Innovation and Employment as part of the NZ SeaRise Programme (Contract ID - RTVU1705) and GNS Science Global Change through Time Programme (Strategic Science Investment Fund, Contract ID - C05X1702), and student grants from the Department of Conservation (Project Number - E4186) and The Nature
Conservancy (Project Number - P120996).

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
