# Peer review of "Characterisation and quantification of organic carbon burial using a multiproxy approach in saltmarshes from Aotearoa New Zealand"

_EGUsphere, 2025_

## Author Comment (AC1)

**Author Response: RC1**

**Dear Reviewers and Associate Editors,**

Thank you for your thoughtful reviews and for allowing us the time to address the comments on our manuscript, "Characterisation and quantification of organic carbon burial using a multiproxy approach in saltmarshes from Aotearoa New Zealand." We appreciate the care taken by the reviewers to strengthen this work.

In response, we have substantially improved the clarity and presentation of the manuscript. Briefly:

- *Structure and readability:* redundant text removed and sections reorganised for clearer flow.
- *Methods clarified and streamlined:* methods streamlined and shortened where appropriate, sampling/processing order, terminology, and statistical approach are now clearly described and consolidated in Methods.
- *Figures corrected and enhanced:* map location fixed; accumulation rates replotted by depth with minor corrections; visuals reorganised for easier comparison and captions clarified.
- *Analyses focused and made more interpretable:* PCA simplified; unnecessary explanations removed; detrending moved to Supplementary; cluster analysis results tabulated.
- *Results/Discussion tightened:* redundant text removed; uncertainty reporting standardised (site mean ± SE); interpretation of $\delta^{13}C$ sources clarified, including limits distinguishing autochthonous vs allochthonous inputs.
- *Practical relevance strengthened:* final section now outlines implications for blue-carbon assessments.

We believe these revisions make the manuscript clearer, more robust, and more useful to the community. Thank you again for your time and constructive guidance. We hope the revised version meets your expectations.

**RC1**

The manuscript entitled *"Characterisation and quantification of organic carbon burial using a multiproxy approach in saltmarshes from Aotearoa New Zealand"*, by Albot et al. presents an impressive body of work, using multiple approaches to characterise and quantify carbon sources, stocks, and accumulation rates in several saltmarsh systems in New Zealand. I want to highlight and commend the authors for the amount of work that went into generating this dataset and preparing the manuscript. A few minor comments are noted below.

There is a mismatch between Figure 1 and the text / supplementary S1: In Figure 1B, there are 10 core locations, but only 9 mentioned in the text. Further, from the supplementary table S1 it seems that Puk-LM4 was omitted in favour of Puk-LM1 or 2, which are located closely together. Is there a reason why this core was excluded and why is it on the figure if not used in the analysis?

**Author response:**

**We thank the reviewer for their comment. Puk-LM4 was a spare sampling location that was never collected and was left on the map by mistake. We have removed this sampling location.**

Lines 139 to 141: You explain methods for DVD in two places [here coma and in line 179] which is confusing. You can combine them in the later section for clarity.

**Author response:**

**This has been removed and included in section 3.2.**

"Dry bulk density (DBD; g cm$^{-3}$) was calculated by dividing the mass of the dry sample by the sample volume following standard methodologies (Howard et al., 2014). Organic carbon density (CD; g C cm$^{-3}$) was calculated by multiplying bulk density by the OC content for each depth interval (wt%; Howard et al., 2014). TOC stocks (Mg C ha$^{-1}$) for each core were calculated by integrating the depth intervals (2, 5 or 10 cm) over the depth range of the core."

Line 179: I assume the DVD was calculated before **freeze-drying**. Otherwise, this sentence is confusing.

**Author response:**

**We have clarified this in Section 3.2.**

"First, large roots and aboveground biomass were manually removed from surface samples to avoid biasing average soil OM properties. We note that samples were not size fractioned as this study focuses on bulk soil OC, which includes belowground living plant biomass (e.g., small rootlets and rhizomes; Macreadie et al., 2017). All samples were then weighed, freeze-dried and weighed again, and homogenised using a ball mill."

Lines 278 to 284: You specify the tests used when normality and equal variance assumptions were not met. What did you use if they were met [or you can specify here if they were all non-normal]? And what was your model structure? Did you test differences across the different sites or any other relationships between variables? Please give more details on the statistical methods.

**Author response:**

**We have rephrased to make it clear that the normality and equal variance assumptions were not met, and therefore, non-parametric tests were conducted on all samples.**

"First, Levene's test for equal variance and Shapiro-Wilk's test for normality were run on all datasets. Because the normality and equal variance assumptions were not met, non-parametric Kruskal-Wallis and post-hoc Dunn's tests were used for pairwise comparisons."

Just a note: It is likely that the variables won't be normally distributed as they are coming from a very diverse set of sites how. However, for an anova it is more important to test for the distribution, heteroscedasticity, and independence of the model residuals rather than the underlying variable distribution.

Lines 302 to 306: There is no need to explain here what a PCA is it overloads the already heavy method section. I suggest removing these lines from the manuscript.

**Author response:**

**Lines 302-306 have been removed.**

Lines 298 to 320: Was PCA performed on all data sets combined or separately? It is not clear from this description what was done here and why. From the results section (lines 526 and following) it seems that PCA was performed on 2 data sets (one without and one with lipids), and on the cores that were age dated (is that correct?). It is unclear as to why these separations were chosen.

**Author response:**

**We have clarified in the Methods that PCA was conducted on the full geochemical dataset (elemental composition, stable isotopes, and XRF), and we report results for two configurations: (i) without lipid biomarker indices and (ii) with lipid biomarker indices, as only a subset of samples had these analyses.**

**To avoid overcomplication and redundancy, we have removed the additional PCA plots restricted to age-dated cores and ran the analyses on the full datasets as you suggest, since these separations did not provide additional insights beyond the full-dataset analyses and added unnecessary complexity. These revisions have been included to improve clarity and focus.**

Lines 351 to 352: This relates to the supplementary Table 2: you should provide more information on the statistical analysis than just the p-values. At least the test statistic in combination with the degrees of freedom should also be included.

**Author response:**

**All test statistics and degrees of freedom have been included with the results in Supplementary Table 2.**

Lines 353 to 354: You don't need to specify the units in the caption if you have them in the table. However, you should explain the acronyms.

**Author response:**

**We have provided explanations for the acronyms in the table caption and removed the units.**

Figure 3: I suggest using a different plot type to visualise the carbon stocks at each site and in the different zones. Bar plots are not meant to visualise continuous data like this. An option could be to plot means and standard error together with the spread of the data [points], to use box plots [with a marker for the mean], or violin or dot plots. This comment has no impact on my recommendation for the article to be accepted but should be seen as advice.

**Author response:**

**We thank the reviewer for this thoughtful suggestion and appreciate the advice on alternative visualisation methods. Our intention was to present a clear and straightforward comparison of average and median carbon stocks between sites and marsh zones, in line with established reporting practices in blue carbon research. While we acknowledge that approaches such as box plots or violin plots can provide additional detail when sample sizes are large, our dataset consists of only three to four discrete measurements per marsh zone. Given this limited number of observations, bar plots offer a straightforward and appropriate way to compare average and median carbon stocks across sites and zones. We have reported the data aligned to methodologies used in previous studies in this field (e.g., Macreadie et al., 2017; McMahon et al., 2023).**

Figure 5: see my comment on bar plops above.

**Author response:**

**We agree that the accumulation rates need to be presented in a different way. We have provided a new Figure 5 showing carbon accumulation rates with depth for each core. During the review and replotting process, we identified and corrected a few discrepancies in recorded DBD calculations across several cores. These adjustments have been incorporated into all datasets and analyses in the revised manuscript. Importantly, these corrections had a minor impact on the updated numbers and do not affect the interpretations or conclusions presented in the study.**

Lines 440 to 480: the equation and explanations of indices should go in their respective methods section. The descriptions of what they indicate may stay in their results if preferred.

**Author response:**

**This has been moved to the methods section.**

Lines 527 and 544: Headings in this section are almost the same. I would suggest changing them to make the differences between the sections easier to grasp.

**Author response:**

**We have updated this section to include data with and without biomarker indices, and to present them in a single section.**

Lines 534 to 536: Do you have documentation on the cluster analysis? It would be informative to show the criteria of the split and the sample groupings in a supplementary table.

**Author response:**

**We have provided the results of cluster analysis p-values in Supplementary Tables 9-12.**

Line 608: Relying too heavily on the mean value here may be misleading, as it is strongly influenced by that one core that hit a peat-bed underneath the sediment layer. It may be better to use the median or at least mention it here. That was also by far the longest core. It is probably worth discussing how this impacts you results (especially since this is the core you used for further analyses).

**Author response:**

**We have clarified this and discuss this in Section 5.1.**

Lines 630 to 632: This sentence explaining why SL rise is unnecessary.

**Author response:**

**We have removed the sentence.**

Lines 662 to 665: Above, you state that C3 plants of terrestrial origin account for a large fraction of the organic material in the marsh sediments but here you say that the marsh plants themselves are responsible for the C3 signature. Does that mean that you are not able to tell the difference between allochthonous terrestrial inputs and autochthonous salt marsh production using the stable carbon isotope tracer? If correct, this limitation should be discussed as well.

**Author response:**

**We have provided the following clarification.**

" Although $\delta^{13}C$ signatures presented in this study typically indicate $C_3$ plant inputs, these values cannot distinguish between saltmarsh vegetation, which is also $C_3$ in NZ, and other terrestrial $C_3$ plants, which include all indigenous NZ plants and at least some of the commercial exotic crops, such as rye grass *Lolium perenne*. Because all cores were collected within saltmarsh sites dominated by saltmarsh species (with surrounding mangroves at all sites except Pāuatahanui) and no/little adjacent forestry or terrestrial vegetation, we interpret the $C_3$ signal as primarily saltmarsh-derived. However, we acknowledge that contributions from transported terrestrial material cannot be excluded as tidal

mixing and river discharge can deliver allochthonous marine- and terrigenous-derived OM (Alongi, 1997; Bulmer et al., 2020; Pondell & Canuel, 2022; Smeaton et al., 2024)."

Lines 768 and following: You used an impressive and resource consuming number of methods to get to the conclusions presented here. I wonder if it would be of benefit to the wider community to comment on a more feasible approach for a scalable solution for blue carbon assessments. E.g., which methods are strictly necessary, and which give important information from a scientific perspective but are less practical to apply?

**Author response:**

**We thank the reviewer for their comments. We have provided the following insights in the last paragraph of Section 5.4.**

"These findings have practical implications for blue carbon assessment methodologies. Current blue carbon methodologies, such as those used by Verra, often require removal of allochthonous material to estimate carbon accumulation rates (Needelman et al., 2018). Results from this and other recent studies (e.g., Li et al., 2025; Peck et al., 2025), suggest that allochthonous inputs can enhance long-term preservation of OC. Furthermore, allochthonous carbon that is transported to restored wetlands in tidally influenced locations is considered "additional" and is excluded from calculations because its preservation results directly from restoration activities (Lovelock et al., 2023a,b). Strict exclusion of allochthonous contribution to saltmarsh OM may underestimate the true sequestration potential of saltmarsh ecosystems. Streamlined protocols that balance scientific rigour with practical requirements are needed to improve carbon mitigation estimates and scale blue carbon projects globally."

Lines 793 to 794: It doesn't make much sense to give uncertainty estimates for a range. I get what you mean (I think), which is that you show the lowest and the highest site means with standard deviation (?) but in that case you should say that. This already threw me in the Abstract where the same case applies.

**Author response:**

**We thank the reviewer for their comment. We reported ±SE for stocks and ±confidence interval ranges for accumulation rates for our maximum and minimum observations, and we agree that this is confusing. We have clarified this in the abstract and conclusions by providing the site mean ± SE for all reported values.**

---

## Author Comment (AC2)

**Dear Reviewers and Associate Editors,**

Thank you for your thoughtful reviews and for allowing us the time to address the comments on our manuscript, "Characterisation and quantification of organic carbon burial using a multiproxy approach in saltmarshes from Aotearoa New Zealand." We appreciate the care taken by the reviewers to strengthen this work.

In response, we have substantially improved the clarity and presentation of the manuscript. Briefly:

- *Structure and readability:* redundant text removed and sections reorganised for clearer flow.
- *Methods clarified and streamlined:* methods streamlined and shortened where appropriate, sampling/processing order, terminology, and statistical approach are now clearly described and consolidated in Methods.
- *Figures corrected and enhanced:* map location fixed; accumulation rates replotted by depth with minor corrections; visuals reorganised for easier comparison and captions clarified.
- *Analyses focused and made more interpretable:* PCA configuration now stated and simplified; unnecessary explanations removed; detrending moved to Supplementary; cluster analysis results tabulated.
- *Results/Discussion tightened:* redundant text removed; uncertainty reporting standardised (mean ± SE); interpretation of $\delta^{13}C$ sources clarified, including limits distinguishing autochthonous vs allochthonous inputs.
- *Practical relevance strengthened:* final section now outlines implications for blue-carbon assessments.

We believe these revisions make the manuscript clearer, more robust, and more useful to the community. Thank you again for your time and constructive guidance. We hope the revised version meets your expectations.

**RC2**

Albot et al., provide a new dataset and exploration of saltmarshes from across New Zealand exploring the quantity and composition of carbon in these system. Saltmarsh carbon data from New Zealand is rare and has been absent from global studies, making this study especially important.

The quantity and quality of the data presented in the manuscript is highly commendable as it could be easily split into several high quality papers. The density of data in the manuscript results in it being hard to follow in places, the is a decision for the authors but splitting this in to two papers would be possible and would not reduce the impact of the research.

Overall the manuscript is written well, but reads more as a thesis chapter opposed to a research article, I would suggest taken time to remove some of the unnecessary information presented.

**Author response:**

**We thank the reviewer for this thoughtful comment and appreciate the recognition of the dataset's value. We have streamlined the manuscript by removing unnecessary detail and improving clarity. While the data could be presented in two papers, we want to present a coherent and comprehensive package of data in 'one place'. This integrated format allows us to present a comprehensive picture of the methods that are available to assess carbon stocks, accumulation rates, and organic matter composition across a range of wetland sites. Our combined approach also allows us to compare and discuss variability in carbon source/provenance and accumulation between sites that have different histories and span several degrees of latitude.**

Introduction

The introduction needs to restructured, though all the information is present in the text, the text is overloaded with unnecessary information that the manuscript does not tackle.

**Author response:**

**We have removed lines 53-72 to make the introduction shorter and have made the writing more concise and to the point.**

Methods

Line 141 - Remove grinder

**Author response:**

**Thank you, this is addressed.**

Line 141 - elaborate on how - Large roots and aboveground biomass were removed
Line 142 - The samples were not size fractioned as this research focuses on bulk soil OC - in the previous sentence it was stated that large roots were removed.
Line 141/142 - Are the sentences ordered correctly did you mill the sample then remove the roots, this seem back to front.

**Author response:**

**Thank you for the comments. We have clarified the sample preparation steps and rationale in the methods section. The revised text now reads:**

"First, large roots and aboveground biomass were manually removed from surface samples to avoid biasing average soil OM properties. We note that samples were not size fractioned as this study focuses on bulk soil OC, which includes belowground living plant biomass (e.g., small rootlets and rhizomes; Macreadie et al., 2017). All samples were then weighed, freeze-dried and weighed again, and homogenised using a ball mill."

**This revision addresses the order of steps and explains why size fractionation was not performed.**

Line 144 - change irMS to IRMS

**Author response:**

**We have changed irMS to IRMS.**

Line 160 - Lead isotope data - state which isotopes

**Author response:**

**We have clarified which isotopes were analysed.**

"For gamma spectrometry, sediment samples were packed into petri dishes and left to equilibrate for three weeks and then analysed to detect radionuclide activity to include $^{210}$Pb, $^{137}$Cs, $^{228}$Ra and $^{226}$Ra (Arias-Ortiz et al., 2018; Goldstein & Stirling, 2003). For alpha spectrometry, the samples were first processed to prepare the granddaughter $^{210}$Po source, and the activity of $^{210}$Po was then measured to calculate excess $^{210}$Pb activities."

Line 160 - 177 - Does slicing the cores at 2cm intervals impact the quality of the radionuclide data, would doing it at 1cm resolution be more useful.

**Author response:**

**We thank the reviewer for their comment. We state that the cores for $^{210}$Pb dating were sampled in 1-cm increments.**

Remove section 3.4

**Author response:**

**We have removed Section 3.4 and integrated some of these methods into Section 3.2.**

Lines 223 - 227 - remove these line they are not required.

**Author response:**

**We have removed these lines.**

Should section 3.6.2 and 3.6.3 be switched as the RPO analysis uses the Py-GCMS data (line 253)

**Author response:**

**We thank the reviewer for this helpful comment. The current order of sections reflects the analytical workflow and the rationale behind it. RPO analysis is performed first to separate $CO_2$ fractions at specific temperature intervals, which are then radiocarbon dated to provide age estimates for different thermal fractions. Py-GC-MS is subsequently applied to separate sample splits to characterise the compound classes within those fractions. While Py-GC-MS can be conducted independently, performing RPO first informs the optimal temperature ranges for the Py-GC-MS ramped heating procedure. This approach ensures that the thermal decomposition steps in Py-GC-MS align with the preservation characteristics identified through RPO, providing deeper insight into the composition and stability of carbon pools. For these reasons, we have retained the sequence where RPO results are presented first, followed by Py-GC-MS analysis.**

Section 299 - you do not need to explain what a PCR is, taken a more direct approach will shorten and improve the manuscript.

**Author response:**

**We have streamlined this section and removed unnecessary text.**

Figure 2 - in the methods you state the troel smith classification scheme is used, can you outline how this aligns with the soil descriptions.

**Author response:**

**We have included an explanation in the Fig. 2 caption that the Troels-Smith classification was simplified to reflect peaty and minerogenic soils. Full descriptions following the Troels-Smith classification system are provided in appendices Fig. S1-S5.**

Section 4.2 - the presentation of the data in the text is not required as it present in table 1.

**Author response:**

**We have removed this section.**

Figure 3 - could the plot be placed side by side.

**Author response:**

**We have placed the plots side by side. During our revisions, we carefully reviewed and replotted all datasets. We identified and corrected minor discrepancies in bulk density calculations for several cores. These corrections have been applied consistently across all datasets and analyses, and the updated values are presented in the revised manuscript. While these adjustments had a minor impact on the results, they do not affect our interpretations or conclusions.**

Figure 4 - the outputs from the Rplum model are not the easiest to understand can you make clear which one was produced by gamma vs alpha spectrometry.

**Author response:**

**We thank the reviewer for this comment. The outputs shown in Figure 4 are generated by the *rplum* model using both gamma and alpha spectrometry data. Specifically, *rplum* integrates measurements from gamma ($^{210}$Pb, $^{226}$Ra, $^{137}$Cs) and alpha ($^{210}$Po) in a single run, so the resulting age-depth models represent a combination of these inputs rather than separate outputs. We have clarified this in the figure caption to make it clear that both gamma and alpha results contribute to the same model.**

Section 4.5 - the use of detrending is interesting but the utility of the method to the manuscript has whole is questionable, I would move this to the sup mat.

**Author response:**

**We agree that detrending was exploratory and have moved this section to the supplementary materials (Fig. S11). To further strengthen the analysis, we re-evaluated the restoration effect using the Okatakata core instead of Awanui. This wetland has a sediment accumulation rate that is closer to the national mean, and reduces the potential bias introduced by the high sedimentation observed at Awanui. We submit that this approach provides a more representative, preliminary assessment of restoration impact.**

Figure 7 - As d15N data has been produced, did improve source estimation.

Section 4.7 - this section does not provide any results, could you remove or provide detail on what was measured.

**Author response:**

**We have revised Section 4.7 to clarify what was measured and how these measurements were used in subsequent analyses. The updated text now explicitly states the elements quantified by XRF. Ratio calculations and proxy interpretations have been moved to the methods.**

Section 4.8.1 - move the equation and ratio discussed to the methodology.

**Author response:**

**We have moved all equations to the methodology.**

Section 4.9 - could the thermograms be displayed.

**Author response:**

**We thank the reviewer for their comment. The thermographs are provided in Figure S9 (Supplementary Materials).**

Discussion

The discussion in generally written well, but as with other section there is a significant amount of unnecessary text, if this could be cut down the manuscript would be much more readable. The above comments concerning the results should inform the discussion.

Again I would like to state that the data in this manuscript is of high quality and the interpolation is well done. However the current structure and dense text struggle to communicate the importance of the study. There is unnecessary text and data that can be removed. I would also ask the authors to consider splitting this into two papers - 1) stocks and accumulation, 2) source

**Author response:**

**Thank you for this valuable feedback and for recognising the quality of the data and interpretation. We agree that clarity and readability are critical for communicating the importance of this study. In response, we have streamlined the manuscript significantly by removing redundant text, condensing overly detailed descriptions, and improving transitions between sections. These changes reduce density and make the narrative easier to follow while retaining all essential information.**

**We considered the suggestion to split the manuscript into two papers; however, we believe that presenting stocks, accumulation, and sources/preservation together provides a more integrated understanding of carbon dynamics at the sites presented in this study.**